# Healthcare-seeking behaviour of fever cases in Magude district, southern Mozambique: A qualitative study

Carlos Eduardo Cuinhane[1,2]☯*, Julia Montaña Lopez[2,3]☯, Hoticha Nhantumbo[2], Helder Djive[2], Ilda Murato[2], Beatriz Galatas[2,3], Caterina Guinovart[3], Francisco Saúte[2], Pedro Aide[4], Neusa Torres[2,5,6], Khátia Munguambe[2,7]

1 Department of Sociology, Faculty of Arts and Social Sciences, Eduardo Mondlane University, Maputo, Mozambique, 2 Centro de Investigação em Saúde de Manhiça, Maputo, Mozambique, 3 ISGlobal, Hospital Clinic—Universitat de Barcelona, Barcelona, Spain, 4 National Institute of Health, Ministry of Health, Maputo, Mozambique, 5 Department of Global Health and Social Medicine, King's College London, London, United Kingdom, 6 SAMRC Developmental Pathways for Health Research Unit, School of Clinical Medicine, University of the Witwatersrand, Johannesburg, South Africa, 7 Faculty of Medicine, Eduardo Mondlane University, Maputo, Mozambique

☯ These authors contributed equally to this work.
* c.cuinhane@hotmail.com

## Abstract

Fever is the main manifestation of malaria, which is a major public health concern in Mozambique. Achieving prompt diagnosis and appropriate management of all malaria cases is key to avoiding severe malaria and death, thus communities need to detect all fevers and seek care promptly. Studies in Magude district have shown that the local population is aware of malaria, including fever as one of the disease symptoms, however, a significant proportion do not seek formal care for fever. This study sought to analyse healthcare-seeking behaviour for fever episodes in Magude and understand its facilitators and barriers to ultimately inform malaria control policies. Using a generic qualitative design, the study included 59 individual semi-structured interviews: 45 with community members (community leaders, elders, adult men and women and teachers) and 14 with healthcare providers. Additionally, 12 focus group discussions with community members were performed. Data were thematically analysed using Nvivo 12 software. The study results revealed that participants recognised fever and categorized it between simple and severe fever. Most participants reported preferring to seek fever care at health facilities, especially for severe fever, but non-formal care sources were also used. The belief that untreated fevers can lead to death and availability and proximity of services and medicines facilitated the use of formal health services, whereas the belief that not all symptoms required formal treatment, and structural barriers (distance, inadequate service delivery and lack of medicines at health facilities) hindered it. In conclusion, healthcare-seeking for fever was an interactive and complex process within and between non-formal and formal

**Data availability statement:** The data of this study were collected under individual-level informed consent after a research protocol was reviewed and approved by CISM's Institutional Ethic Committee (CIBS-CISM) and the Mozambican Ministry of Health National Bioethics Committee. The informed consent signed by the participants stated that: "data will only be available for the study team", and the protocol established that all information will be confidential, and no data from data collection forms, nor from audio files will be accessible to anyone outside of CISM. Given this statement approved by the IRBS, data from this study is available upon request to these institutional review boards: CISM's institutional Ethic Committee (secretariado.cibis@manhica.net) or the Mozambican Ministry of Health National Bioethics Committee (cnbs.mocambique@gmail.com) for researchers who meet the criteria for access to confidential data.

**Funding:** The study reported in this paper was part of the Magude Project. The Magude project (NCT02914145) was funded by the Bill and Melinda Gates Foundation and Obra Social "la Caixa" Partnership for the elimination of malaria in southern Mozambique (OPP1115265). The Magude project was implemented by Centro de Investigacao em Saude da Manhica (CISM). CISM is supported by the government of Mozambique and the Spanish Agency for International development (AECID) for core funding. But the study reported in this paper did not receive any funding from the government of Mozambique nor AECID. It was fully and solely funded by the OPP1115265. The funders have no role in study design, data collection, and data analysis, decision to publish and for the preparation of the manuscript.

**Competing interests:** We have read the Journal's policy and the authors of this manuscript have the following competing interest: CEC, HN, JML, HD, IM, FS, PA, NT, KM are paid employees of the Centro de Investigacao da Manhica (CISM), which is supported by the Government of Mozambique and Spanish Agency for International development (AECID). The salary of CEC, JML, JML, HD, IM, PA and KM were partially or fully covered through the Magude MALTEM project, which is co-funded by the Bill and Melinda Gates and Obra Social "la Caixa". However, this does not alter our

health services, influenced by structural, community and individual factors. Malaria control and elimination strategies should simultaneously address these by improving the quality and accessibility of formal health services and sensitizing the community of the importance to seek formal health care for all fever severities.

## Introduction

Fever [1] is considered the main manifestation of malaria [2–6], which is a major public health concern in endemic areas. Globally, there were an estimated 249 million malaria cases in 2022, out of which 233 million (94%) were in Africa [7], and Mozambique accounted for 4% of all worldwide malaria cases [5]. Malaria control continues to encounter several challenges, with low coverage and effectiveness of preventive measures and appropriate case management [8,9], due to weak health systems [10], infrastructural factors, climate and environmental factors [11], economic factors [11,12], delays in treatment seeking [13,14], fear of side effects of malaria treatment drugs, limited access to health education about malaria disease and its treatment [15], misdiagnosis and missed diagnosis [11], low accessibility of health facilities [15], drug and insecticide resistance, and decision making [12].

Seeking prompt care for fever, the main malaria symptom [6], to receive adequate diagnosis and treatment is a cornerstone of malaria control [6,16,17]. Other manifestations include chills, sweating, headache, nausea, muscle pain and vomiting [11]. Moreover, there is a geographical overlap between high malaria transmission and high prevalence of fever [5,7]. Between 2015 and 2022, the prevalence of fever in sub-Saharan Africa was 23%, but only about 66% sought treatment, of which 69% received care from public health facilities, 2% from community health workers, and about 40% received care from both the formal and informal private sectors [7]. Moreover, studies have indicated that a proportion of people rarely seek care for fever cases in public health facilities [18].

In Mozambique, a national malaria survey conducted in 2018 indicated that the prevalence of fever was 18%, of which 69% sought care. Of these, 96% received care from a public health facility, 2% from the private health sector and 2% from the informal sector, such as markets, traditional healers and parents or friends [19]. On the other hand, the latest national Demographic and Health Survey (2022–2023) showed that malaria is the leading cause of under-five mortality, with 32.3% of children infected by malaria parasites [20]. While the southern Maputo province has reached pre-elimination levels with an under-five malaria prevalence of 0.3%, the centre and north of the country are severely affected by the disease, with the highest burden found in the northern Nampula province, with 54.7% prevalence [20]. Thus, seeking care for fever or any other malaria symptom early on is crucial for preventing severe malaria and death. To reverse the current malaria situation, the country has been implementing measures to accelerate malaria control and elimination, including the distribution of long-lasting insecticidal nets (LLINs), indoor residual spraying (IRS), continuous community engagement campaigns to increase health education

adherence to PLoS ONE Policies on sharing data and materials. There is no patent, product in development or marketed product associated with this research to declare.

about fever, malaria and its treatment to increase health-seeking behaviour for fever, and strengthening of case management and the surveillance system [21–23].

The Magude district, in Maputo province, benefited from a pilot malaria elimination program led by the Manhiça Health Research Centre (CISM) in collaboration with the National Malaria Control Programme (NMCP) from 2015 to 2020. The program consisted of implementing a comprehensive combination of interventions including LLINs, IRS and two yearly rounds of mass drug administration (MDA) in 2015 and 2017 [24], accompanied by continuous community engagement campaigns, incentivising the population to adhere to and use malaria control tools and to seek prompt care for fever from health facilities or community health workers (CHWs) [25,26]. After achieving a drop in malaria transmission from moderate to very low following the last MDA rounds, a reactive surveillance strategy was implemented in Magude from July 2017 until January 2020 to maintain the gains, as recommended by the World Health Organization [27]. With the reactive surveillance strategy being triggered by the detection of malaria cases at health facilities and by CHWs, the success of the intervention and the progress towards elimination highly depends on whether community members are able to detect fever and seek care promptly from the formal health sector. The current article reports qualitative results about fever healthcare-seeking behaviour and decision-making processes in Magude district to inform malaria control policies.

Previous studies [28] in this district revealed that the local population was aware of malaria and its presentation, including fever as one of the disease symptoms. "Fever" was sometimes used generically to describe the general feeling during an episode of malaria, but there was a general understanding of the distinction between fever and malaria [28]. Thus, it is expected that, when facing a probable malaria episode, the feeling of fever could be a key trigger for healthcare seeking. However, it has been registered that a considerable number of people still do not seek healthcare when experiencing fever, as evidenced by a cross-sectional survey in the Magude district [23] which found that between 30% and 40% of surveyed individuals with fever reported not seeking care at formal healthcare settings. This is in line with a national survey about malaria indicators, undertaken in 2018, which also indicated that about 40% of people living in rural areas and about one third of all the surveyed population did not seek healthcare when they experienced fever [19].

The literature has documented several factors hindering healthcare-seeking behaviour among fever cases in low-income countries. In the sub-Saharan African literature, long distances to health services seems to be the main barrier to seeking healthcare in Malawi [14], Tanzania [29] and Madagascar [30], and poverty in all across sub-Saharan Africa [31]. Lack of economic resources to get to health facilities and to buy medicines was also found as a barrier to seek healthcare in Kenya [32], Malawi [33] and other sub-Saharan African countries [34]. Likewise, low educational level of caregivers was an important barrier to seek healthcare in Tanzania [29] and other sub-Saharan African countries [34], while living in rural areas [33] and lack of women empowerment to take decisions to seek healthcare were considered the main blockage in Malawi [14]. Moreover, fear of mistreatment from healthcare providers

[30] or fear of receiving inadequate care was also a common barrier, coupled with the unavailability of antimalarial medicines [14,30]. Preference for home remedies and traditional healers also hindered healthcare-seeking in Malawi [14] and Madagascar [30]. The low perceived value-for-money or value-for-time of seeking healthcare fuelled the preference for home remedies and traditional healing in Malawi [14], and this seemed to be substantiated by a perceived shortage of staff in health facilities and inconvenient opening times [14]. In Mozambique, previous studies show that barriers preventing fever cases from seeking healthcare include poverty [29,35], long distance to access health services and lack of medicines at the health facilities [23,35,36], low educational level [36] and perception of low-quality services provided by the health professionals [23].

As the district of Magude and the wider Maputo province in southern Mozambique move towards malaria pre-elimination levels in line with the Malaria Elimination Initiative ambition to eliminate malaria in eight southern African countries, it becomes ever more necessary to walk the last mile and identify the remaining pockets of malaria transmission. Identifying and acting on all malaria cases is currently not possible because a portion of the population do not access formal health services when they have a fever [24]. Understanding the rationality and the behaviour of the population in relation to fever and overcoming the barriers they may encounter to accessing formal healthcare is therefore a necessary and instrumental step in order to attain a desirable coverage of malaria case detection and management. Previous studies [23,28] about malaria and its perceived symptoms in the district were not exhaustive about the healthcare-seeking behaviour of the population, and little is known about the rationality and decision-making processes regarding healthcare-seeking when people experience fever. This study, therefore, sought to analyse the healthcare-seeking behaviour during fever episodes and identify facilitators and barriers to seek care within the available formal health services in the district of Magude, using the social ecological model [37]. In this study, this model is used to identify and analyse multiple factors influencing health-seeking behaviour during fever episodes.

## Methods

### Study site

Magude district is located in the northwest of Maputo province, southern Mozambique, with 63,691 inhabitants and 14,583 households registered in the most recent national census in 2017 [38]. It has five Administrative Posts: Magude village, Motaze, Mahel, Panjane and Mapulanguene [39], and all five are covered by the study. The level of malaria in the district is considered as very low malaria transmission (<100 annual cases per 1000 population) after the implementation of a set of interventions aiming for elimination of malaria in the district [23].

The district has nine formal public rural health facilities, one referral main public health centre located in Magude village [24,40]. The main referral centre in Magude village attends approximately 60% of all outpatient department visits in the district, followed by Motaze's (16%) and Moine's (7%) health facilities, while the other health facilities attend less than 5% of patients each [23]. The district has three medical doctors, 15 general nurses and 19 maternal and child health nurses [24,40], which together result in a ratio of 0.58 health workers per 1000 population. The median Euclidean distance between households and their nearest health facility is 2.7km, with a range between 15m as the closest to 38km as the furthest distance from a household to the closest health facility [24]. Additionally, the district is served by 27 community health workers (CHWs), who provide diagnosis and treatment of malaria and other diseases such as diarrhoea and pneumonia, and refer patients with signs of other illnesses requiring medical attention to health facilities [41]. These CHWs are considered formal. Each CHW reports to a health facility and offers healthcare services in collaboration with health facilities. The average distance between households and the nearest CHW is 6.3 km and the median distance is 4.4 km [24]. Both health facility workers and CHWs engage in sensitization of the community about malaria using a social behaviour change communication approach oriented by the National Malaria Control Programme (NMCP) [42], and they offer healthcare services almost for free to all patients (1 metical (USD 0,016) per adult consultation and free for all children). Apart from these formal healthcare services (health facilities and CHWs),

there are also formal private pharmacies, and informal healthcare services provided by traditional healers, informal salespersons selling medicines, and other community members with knowledge of ethnomedicine who may advise or provide healthcare services within the family level.

## Study design

This generic qualitative research study [43–45] is nested within the broader Magude malaria elimination pilot project [26], which included four cross-sectional surveys [23] conducted to evaluate the malaria prevalence level in the district of Magude on an annual basis (2015–2018). These surveys also quantified the proportion of participants who sought care within the formal healthcare system for a fever episode and participants' timing of healthcare-seeking, and revealed that a considerable number of people were not seeking care in the available formal health facilities during fever episodes [26]. Thus, this qualitative generic study sought to analyse the perspectives and perceptions of the community about fever as well as processes, facilitators and barriers of seeking treatment care in the available formal health facilities among inhabitants of the Magude district.

## Study participants

The study included residents of Magude with different social roles and characteristics who were selected to capture a range of perceptions about health-seeking behaviour for fever cases. The study participants categories included community leaders, household heads, women with decision-making power who were mothers of children under 15 years of age (defined as those who were able to independently decide to seek healthcare upon health issues regardless of their marital status), women without decision-making power who were mothers of children under 15 years of age (considered as those whose decision to seek healthcare laid on their family members), pregnant women, elders over 60 years old who were or not caregivers of children under 15 years of age, travellers (defined as participants who had slept outside their homes in the last 30 days), teachers, and different categories of health providers, including traditional healers, community health workers (CHWs) and health professionals working at health facilities. These categories were chosen in order to include a maximum variation of the study sampling, following the recommendations of qualitative generic study [44]. Moreover, the study sought to access the perceptions of different groups, comprising all known health providers and members of the community as potential users of health services and members of the community who had decision-making about when and where to seek care for fever cases.

## Sampling

The study used purposive sampling to select all study participants. The sampling method was based on various sampling frames. Data collected during the annual cross-sectional survey undertaken between September 2017 and June 2018 [23] was used to categorise different individuals living in Magude with the following specific characteristics: pregnant women, travelling people and general population. Pregnant women and travelling people were identified with the help of community leaders, purposively selected and included in the study sample. The malaria elimination project [26] collected information on the figures in the district, from which community leaders, health professionals and CHWs were identified for purposive recruitment in the study. Community leaders also collaborated in the identification of individuals with specific characteristics: household heads, women with or without decision-making power who were mothers of children under 15 years old, elders with or without a caregiving role, and traditional healers who were registered in the Association of Traditional Healers of Mozambique (AMETRAMO). To confirm the categories of women, the researcher asked the selected women whether they had decision-making power in health-seeking about health issues or not before the start of the interview. Teachers were identified and recruited in the local primary schools. Apart from characteristics of each study group, other inclusion criteria were: living in the Magude district, being older than 18 years of age, being able to provide informed

consent, expressing willingness to participate in the study in a written informed consent form, while the exclusion criteria were being sick or not willing to participate in the study.

The study included a total of 59 semi-structured interviews (SSIs), out of which 45 were with members of the community and 14 with healthcare providers, and 12 focus group discussion (FGDs) with members of the community. This sample was determined by the saturation theory [46]. Based on this theory, researchers stopped to include more participants in the study when they noticed that no new data or themes were emerging during interviews. The sampling of participants per administrative post in SSIs and FGDs is shown in Table 1.

## Procedures

Semi-structured interviews (SSIs) and focus group discussions (FGD) were carried out between 4th September 2019 and 30th January 2020. SSIs and FGD guides were developed for data collection. A total of five SSI guides were designed to collect data from different population groups. One general interview guide was developed and used in interviews with household heads, women who were mothers of under 15 years old with or without decision-making power, pregnant women, elders and community leaders. The general interview guide focused on perceptions of illness, perceptions and causes of fever and healthcare-seeking behaviour for fever cases, and facilitators and barriers to seek healthcare at formal healthcare services (health facilities and CHWs) for fever cases. Other specific interview guides were designed for each of the remaining groups: travellers, teachers, traditional healers, CHWs and healthcare professionals. The specific interview guides explored additional issues related to the attitudes and the specific role of the participants belonging to each of these groups in relation to fever cases. A single general FGD guide was developed to explore perceptions of general illness and fever, healthcare seeking-behaviour for fever cases, and facilitators and barriers to seek healthcare for fever cases at the available formal services in the study setting. The FGD guide and the general interview guides were pre-tested in the Xinavane community (a neighbouring community adjacent to the study setting).

All healthcare professionals, CHWs and teachers were interviewed in Portuguese, while interviews with the community members were conducted in local languages (Changana) or in Portuguese, depending on the language that each participant felt most comfortable with. Interviews among community groups lasted between 40 and 60 minutes, while the duration of interviews with healthcare providers varied between 20 and 40 minutes. All FGDs were conducted in local language (Changana), and lasted between 60 and 90 minutes. Three social scientist researchers, namely two females and one male, collected the data. The researchers were under supervision of a female lead social scientist research coordinator. All researchers conducted both semi-structured interviews and FGDs. Community members were interviewed in their households and FGDs took place in the community, while interviews with healthcare providers took place in the health facility where they work.

All interviews and FGDs were audio recorded following the consent of the participants. Some notes were taken during the interview and completed immediately after the interview was finalized in the format of field notes. Each FGD was conducted by two researchers: one was the moderator and the other one was the note taker of the non-verbal behaviours and dynamics of the discussion while ensuring recording was in place. The lead social scientist researcher together with the other two researchers reviewed the audio recordings of each SSI and FGD and held briefing sessions to discuss the findings and limitations before performing further interviews and FGDs to ensure data quality, address possible gaps identified during data collection, and to monitor saturation of the categories.

## Data analysis

All audio recorded data were independently transcribed and the quality and accuracy of the transcriptions was controlled for by contrasting the transcriptions with the audio. Researchers thoroughly read the transcriptions and preliminarily coded them according to a discussed and approved preliminary generic outline of nodes written in the qualitative data analysis software Nvivo 12. A thematic analysis [47] was used to define the themes, mostly pre-determined prior to data

**Table 1. Distribution of participants according to data collection procedure and location within the district of Magude.**

| Administrative post | SSI with community members | | | | | | | | | SSI with health providers | | | | FGD | | | | |
|---|---|---|---|---|---|---|---|---|---|---|---|---|---|---|---|---|---|---|
| | Community leaders | Household heads | Elders who were not caregivers | Women with decision-making | Women without decision-making | Pregnant women | Teachers | Travelling people | Total | Traditional healers | CHWs | Healthcare professionals | Total | Household head | Women with decision-making | Women without decision-making power | Elders who were caregivers of children | Total |
| Magude village | 1 | 1 | 1 | 2 | 5 | 2 | 1 | 3 | 16 | 1 | 1 | 1 | 3 | 1 | 0 | 1 | 0 | 2 |
| Motaze | 1 | 1 | 1 | 3 | 5 | 3 | 1 | 2 | 17 | 1 | 1 | 1 | 3 | 1 | 0 | 1 | 1 | 3 |
| Panjane | 0 | 1 | 1 | 0 | 2 | 0 | 1 | 0 | 5 | 1 | 1 | 1 | 3 | 0 | 1 | 1 | 1 | 3 |
| Mahel | 0 | 0 | 1 | 0 | 0 | 0 | 1 | 0 | 2 | 1 | 0 | 1 | 2 | 1 | 0 | 1 | 0 | 2 |
| Mapulanguene | 2 | 0 | 1 | 1 | 0 | 0 | 1 | 0 | 5 | 1 | 1 | 1 | 3 | 1 | 0 | 0 | 1 | 2 |
| Total | 4 | 3 | 5 | 6 | 12 | 5 | 5 | 5 | 45 | 5 | 4 | 5 | 14 | 4 | 1 | 4 | 3 | 12 |

capture but also incorporating themes emerging from the data collected. The identified themes and subthemes were discussed among the research team until a consensus was reached. The generated final themes were: illness identification and interpretation of signs and symptoms, healthcare-seeking behaviour of fever cases, and facilitators and barriers to health-seeking in CHWs and health facilities for fever cases. Most subthemes branching out of the main themes emerged from the data collected.

## Ethical considerations

The study obtained ethical clearance from the CISM Institutional Bioethics for Health Committee [*Comité Institucional de Bioética para a Saúde do CISM*] (CIBS-CISM), protocol number: Ref CIBS-CISM/041/2019. Written information about the objectives, procedures, risks and benefits of the study was provided to participants, alongside an oral explanation of the contents of the participant information sheet. Written, informed consent was obtained from all the participants. Participants were assured about their anonymity and confidentiality throughout the research process. Thus, all participants names were not recorded, and all informed consents, digital records and databases were securely stored at a CISM server.

## Results

The study completed 59 semi-structured interviews (SSIs), 45 of which were with members of the community and 14 with healthcare providers, and 12 focus group discussion (FGDs) with a total of 147 members of the community. Among community members, the majority were female, living in marital union, had some formal education, and were famers. Moreover, most women with decision-making power were widows, while most of those women without decision-making power were in marital union (Tables 2 and 3).

Among formal healthcare providers (CHWs and health facility-based professionals), 5 were male and 4 were female, 7 had secondary education and had experience on performing their duties, while among traditional healers, 4 were female and 1 was male, with some formal education and more than 16 or more years of experience on performing their duties (Table 4).

### Thematic table of findings

This study explored four themes and several sub-themes emerging from the data. The first theme analysed illness identification and interpretation of signs and symptoms, and it included sub-themes, such as illness recognition, concept and interpretation of fever and perceived signs and causes of fever. The second theme described healthcare-seeking behaviour of fever cases, including the following sub-themes: decision-making and timing to seek for care, healthcare-seeking at health facility, healthcare-seeking at community health workers, healthcare-seeking at pharmacies and informal salespersons, healthcare-seeking at traditional healers, and self-treatment at home. The third theme identified perceived facilitators to seek care for fever within the available formal health services, and the fourth theme identified perceived barriers to seek care for fever within the available formal health services in Magude district (Table 5).

### Illness identification and interpretation of signs and symptoms

**Illness recognition.** Participants belonging to all groups, regardless of their place of residence, reported that they were able to recognize when children and adults were ill. They used direct observation (evaluation of physical signs) combined with their everyday experience of living and interacting with people to identify the health state of a person. Participants assumed that the child was undergoing certain symptoms such as pain based on physical signs including feverish body, warm body, pale body, red eyes, difficulty in breathing, body weakness, body thinness, loss of skin brilliance, feeling cold and vomiting. Behavioural signs were also used to identify illness, including isolation, unhappiness, not playing or talking as usual, lack of appetite, as well as crying and not sleeping as usual. The same physical and behavioural signs were also used to identify illness among adult people.

*"I can identify that the child is not well, because normally she/he plays, but when she/he is not playing, she/he is just crying, so I put my hand on her/his head and I feel that she/he is getting warm, the child is not well; that is when I take her/him to the health facility before the illness gets worse"* (SSI 05, pregnant woman, Motaze).

**The concept and interpretation of fever.** In the local language, Changana, fever was referred to as *dzedzedze* or *muzothotho* in the Magude Village, Motaze, Mahel and Mapulanguene communities, and as *mututumela* in the Panjane community. Both *dzedzedze or muzothotho* and *mututumela* were characterized by the same set of symptoms linked to fever, such as body warming, loss of strength and feeling cold.

*"It's when your body heats up (…); dzedzedze, it's an outbreak that sometimes you don't even understand where it comes from and it attacks you, you can find the person in the daytime sleeping, covered with blankets, feeling cold, but it's sunny, and they say it's a fever (…)"* (FGD 02- men, household heads, Motaze).

Participants categorized fever into two categories. The first category was "simple fever" linked to headache, coughing, shivering and/or feeling cold, which they believed could be resolved and disappear in a short time. The second category

**Table 2. Sociodemographic characteristics of SSIs participants.**

| Variables | Frequency (n) | Percentage (%) |
|---|---|---|
| **Sex** | | |
| Male | 13 | 29 |
| Female | 32 | 71 |
| **Age (Years)** | | |
| 19–35 | 25 | 56 |
| 36–49 | 9 | 20 |
| 50–59 | 4 | 9 |
| 60 + | 7 | 15 |
| **Marital Status** | | |
| Single | 9 | 20 |
| Married | 4 | 9 |
| Union (Marital Union) | 26 | 58 |
| Widowhood | 6 | 13 |
| **Education** | | |
| None | 9 | 20 |
| Primary | 20 | 45 |
| Secondary | 15 | 33 |
| High Education | 1 | 2 |
| **Occupation** | | |
| Farmer | 32 | 71 |
| Service/Labourer | 4 | 9 |
| Student | 1 | 2 |
| Teacher | 5 | 11 |
| Salesperson | 3 | 7 |
| **Religion** | | |
| Christianity | 39 | 87 |
| Animism | 5 | 11 |
| Atheism | 1 | 2 |

was "severe fever" accompanied by sweating, fatigue, vomiting and/or diarrhoea. They perceived that the second type of fever could indicate several illnesses, including malaria, but only health professionals could identify the specific illnesses following a diagnosis procedure.

"[There are several types of fevers, such as…] *Headache... you ask how is the child, and he replies that he doesn't feel well, he is sick, he is warming up, while it is headaches that afflict him. Another haaa... has diarrhoea... here we only know that it is fever, but only doctors after testing can diagnose the type of the fever*" (FGD 01, elders who were caregivers of young children less than 15 years old, Panjane).

"(…) *When you feel body aches, headache, shaking, then they say you have dzedzedze (…) you have to do a test, because it could be malaria. It has various forms. There is a cold, that's when I can't wake up, and there is a simple cold, which disappears in a short time after starting (…)*" (SSI 02, community leader, Panjane).

Regardless of the type or category of fever, the participants' discourses revealed that it was an indication of an underlying illnesses such as malaria, respiratory or gastroenteric problems mainly. However, several participants associated fever to malaria, as one of the participants highlighted in the following discourse.

**Table 3. Sociodemographic characteristics of FGDs participants.**

| Variables | Categories of participants | | | |
|---|---|---|---|---|
| | Household head (n = 43) | Elders caregivers of children (n = 32) | Women with decision-making power (n = 24) | Women without decision-making power (n = 48) |
| **Sex** | | | | |
| Male | 43 (100%) | 7 (22%) | 0 (0%) | 0 (0%) |
| Female | 0(0%) | 25 (78%) | 24 (100%) | 48 (100%) |
| **Age (years)** | | | | |
| 19–35 | 11 (26%) | 0 (0%) | 9 (37%) | 28 (58%) |
| 36–49 | 8 (19%) | 0(0%) | 6 (25%) | 11 (23%) |
| 50–59 | 14 (32%) | 0 (0%) | 5 (21%) | 5 (11%) |
| 60 + | 10 (23%) | 32 (100%) | 4 (17%) | 4 (8%) |
| **Marital Status** | | | | |
| Single | 1 (2%) | 0 (0%) | 2 (8%) | 5 (10,5%)) |
| Married | 3 (7%) | 1 (3%) | 0 (0%) | 4 (8%) |
| Union (Marital Union) | 36 (84%) | 13 (41%) | 14 (59%) | 34 (71%) |
| Widowhood | 3 (7%) | 18 (56%) | 8 (33%) | 5 (10,5%) |
| **Education** | | | | |
| None | 7 (16%) | 26 (81%) | 14 (58%) | 8 (17%) |
| Primary | 33 (77%) | 5 (16%) | 9 (38%) | 36 (75%) |
| Secondary | 3 (7%) | 1 (3%) | 1 (4%) | 4 (8%) |
| **Occupation** | | | | |
| Farmer | 26 (60%) | 26 (81%) | 24 (100%) | 35 (73%) |
| Service/Laborer | 3 (7%) | 0 (0%) | 0 (0%) | 6 (12%) |
| Salesperson | 12 (28%) | 6 (19%) | 0 (0%) | 7 (15%) |
| Retired | 2 (5%) | 0 (0%) | 0 (0%) | 0 (0%) |
| **Religion** | | | | |
| Christianity | 27 (63%) | 21 (65%) | 15 (62%) | 44 (92%) |
| Animism | 0 (0%) | 5 (16%) | 0 (0%) | 0 (0%) |
| Atheism | 16 (37%) | 6 (19%) | 9 (38%) | 4 (8%) |

*"Yes, I know what is fever. Fever is when you feel pain in the body, headache, let's say they are brothers with malaria"* (SSI 02, man, household head, Magude village).

**Perceived signs and causes of fever.** Several community members, regardless of their place of residence, used physical and behavioural signs to identify symptoms linked to fever in both adults and children. Physical signs and symptoms, such as body pain, body warmth or high body temperature, coughing, vomiting, red eyes, shaking, stomachache, feeling cold, headache and pale body in both adults and children were perceived as indicators of fever. Behavioural signs, such as loss of appetite among children and adults, and children not playing as usual, getting bored and crying were also perceived as indicative of fever. However, some women both with and without decision-making power, some elders who were not caregivers and some household heads said that they were not able to identify signs or symptoms linked to fever specifically.

**Table 4. Sociodemographic characteristics of SSI among healthcare providers.**

| Variables | Categories of participants | | | Total |
|---|---|---|---|---|
| | Health professionals in health facilities (n=5) | Community health workers (n=4) | Traditional healers (n=5) | |
| **Sex** | | | | |
| Male | 2 | 3 | 1 | 6 |
| Female | 3 | 1 | 4 | 8 |
| **Age (Years)** | | | | |
| 19–35 | 4 | 2 | 0 | 6 |
| 36–49 | 1 | 2 | 0 | 3 |
| 50–59 | 0 | 0 | 0 | 0 |
| 60 + | 0 | 0 | 5 | 5 |
| **Marital Status** | | | | |
| Single | 4 | 1 | 2 | 7 |
| Married | 0 | 0 | 1 | 1 |
| Union (Marital Union) | 1 | 3 | 0 | 4 |
| Widowhood | 0 | 0 | 2 | 2 |
| **Education** | | | | |
| None | 0 | 0 | 1 | 1 |
| Primary | 0 | 2 | 4 | 6 |
| Secondary | 5 | 2 | 0 | 7 |
| **Occupation** | | | | |
| Nursing | 5 | 0 | 0 | 5 |
| CHW | 0 | 4 | 0 | 4 |
| Traditional medicine | 0 | 0 | 5 | 5 |
| **Work experience** | | | | |
| 1-5 years | 4 | 0 | 0 | 4 |
| 6-10 years | 0 | 2 | 1 | 3 |
| 11-15 years | 1 | 1 | 0 | 2 |
| 16 years + | 0 | 1 | 4 | 5 |
| **Religion** | | | | |
| Christianity | 5 | 4 | 2 | 11 |
| Animism | 0 | 0 | 1 | 1 |
| Atheism | 0 | 0 | 2 | 2 |

**Table 5. Thematic table of study findings.**

| Themes | Sub-themes | Description of content |
|---|---|---|
| **Theme 1:** Illness identification and interpretation of illness signs and symptoms | Illness recognition | Physical and behavioural signs used to identify the health status of a family member |
| | Concept and interpretation of fever | Local terminology for feverish states and their classification |
| | Perceived signs and symptoms of fever | Signs used to identify fever symptoms |
| **Theme 2:** Health care seeking behaviour | Decision-making and timing to seek care for fever cases | Actors with power to decide when and where to seek care. Reasons to delay or seek healthcare immediately after the onset of the fever |
| | Healthcare-seeking at health facilities | Reasons and preferences o seek care at formal health facilities, Perceived types of diseases and fever treated at health facilities. |
| | Healthcare-seeking at community health workers | Reasons to seek community health workers; Perceived types of illnesses and fever treated at community health workers. |
| | Healthcare-seeking at pharmacies and informal drug salespersons | Reasons for using self-medication; Type of illnesses and fever susceptible to self-treatment using drugs from pharmacies and informal salespersons |
| | Healthcare-seeking at the traditional healers | Perceived role of traditional healers in fever treatment. |
| | Self-treatment at home | Reasons for self-treatment using home-based remedies; Type of illnesses and fever susceptible to self-treatment at home. |
| **Theme 3:** Perceived facilitators to seek care for fever within the formal health services | | Structural, contextual (community) and individual factors that enable the rapid seeking of care for fever cases. |
| **Theme 4:** Barriers to seek care for fever within the formal health services | | Structural, contextual (community and health facility level), and individual factors that hinder the rapid seeking of care |

"*Disease has various ways of entering a person, to the point of realising that it is fever, you may find someone sitting feeling cold, freezing, even if you feel heat, he feels cold, that is fever. You have other people who feel warmness, while you feel cold, that's the fever coming in (…)*" (SSI 05, community leader, Mapulanguene).

"*If he is a child, when he has fevers, he can't play, can't eat, gets hot... in the afternoon he gets hot, from one time to another he is well, from one time to another he gets hot*" (FGD 01, women without decision-making power, Panjane).

"*Participant: Fever? I don't know, but if it comes on, I might not stay here, I can run to the health facility, so they can see what it is.*

"*Researcher: How can you identify that this illness is fever?*.

*Participant: I don't know, I cannot know, in the child I can only see that this child is sick. If he is not well, then it is the doctors who will know there*" (SSI 01, woman without decision-making power, Magude village).

In contrast to the discourses of participants who reported being able to identify fever and its signs, CHWs considered that most members of the community were not able to identify fever, and all they knew was that children or adults themselves were ill. They added that at most they would interpret the fever episode as malaria, leaving it up to the CHW to confirm or discard it.

Most participants identified mosquito bites as one of the main causes of fever. Other perceived causes of fever were poor home environmental hygiene, such as the lack of proper cleaning of the house and the yard, the lack of adequate toilets and the inadequate use of toilets, poor personal hygiene related to the lack of hand washing before meals and after

toilet use, consumption of inadequate water, dust, exposure to the sun or cold, lack of water in people's body and playing under the rain or in stagnant dirty water.

> "*What causes the fever is the grass, if you don't sweep the yard, when there is stagnant water, that water proliferates mosquitoes, and the bush proliferates mosquitoes, that's what causes the fever. If you are bitten by malaria mosquitoes, it can cause the fever*" (FGD 03, women without decision-making power, Motaze).

> "*Fever comes in different ways, it happens, this wind that is blowing, raising dust and entering through the nostrils, causes unnecessary coughing, and when you cough, that means it is fever, because you caught the dust that entered through the nostrils and you have no protection, you catch the cough*" (FGD 02, men, household heads, Motaze).

However, some participants, particularly some women without decision-making power, elders who were not caregivers of children and pregnant women reported that they did not know the causes of fever, but they recognized that fever was an illness. On the other hand, teachers, CHWs and health professionals predicted that the main causes of fever in Magude district was malaria caught through the bite of the anopheles' mosquito. Health professionals, in particular, indicated that fever was also caused by pneumonia, respiratory tract infections and *dermatitis bacterianas* (dermatitis bacteria).

> "(...) *There are people who do not use the mosquito nets properly, while they have them, and when it comes to the mosquito, that anopheles which is the one that causes malaria, then it can be the cause of fevers*" (SSI 05, teacher, Panjane).

> "*Well, here in our area, the main causes of fevers have been upper respiratory tract infections (...), we also have cases of malaria due to our region, (...) malaria is an endemic disease in our country, so in addition to malaria we can also find pneumonia, we can also find some skin diseases, namely bacterial dermatitis, which can break out with fever*" (SSI 03, health professional, Magude village).

CHWs and health professionals perceived that most members of the community were able to identify fever as sign of malaria because they know malaria symptoms. Aligned with this perception, almost all participants regardless their place of residence, including those who reported that did not know the cause of fever, correctly identified malaria collective symptoms as fever, headache, body pain, joints pain, vomiting, lack of appetite, diarrhoea, and all indicated mosquitoes as the main cause of malaria.

### Healthcare-seeking behaviours

Participants reported seeking healthcare to different places when they perceived different illnesses, including health facilities, CHWs, private pharmacies and informal drug sellers, self-treatment at home, traditional healers, or combinations of the aforementioned sources. Participants used direct observation of physical signs and symptoms, their previous experiences and the evaluation of the feasibility of available options to assess and decide upon the type of care to seek. The following analysis presents different trajectories of healthcare-seeking following the identification of general illnesses and fever by study participants.

**Decision-making and timing to seek care for fever cases.** The decision-making processes in relation to when and where to seek healthcare varied from family to family, but collectively could be summarized into a few categories: in some families headed by women or where the husband was absent, women were the main decision-makers about healthcare-seeking upon illness of their children and/or themselves as part of their caretaking role in the family, and they generally informed other family members (mainly their husbands or other family members living

with them) about the course of action they were to pursue. In some cases, especially when husbands lived away from the family, women took decisions about healthcare-seeking, and informed the husband after coming back from the health facility. Men were generally always the decision-makers about seeking care for themselves, and there were no non-heterosexual family configurations included in this study. In some families headed by men, decision-making on when and where to seek healthcare relayed on more than one family member, generally on both husband and wife. Several participants said that the husband or both the husband and the wife took the decision about when and where to seek healthcare for fever cases. But in the absence of a man as household head, the wife took the decision, or she informed other members of the family such as in-laws, who took the decision. In other families also headed by men, the male household head was the main decision-maker and the women had to obtain authorization from the husband to take the child to the health facility when sick, except from when the husband was absent, in this case women could decide. The latter case was mostly reported either by male household heads or by women without decision-making power.

> "*Participant 3: Yes, it is the husband who decides, sometimes the child feels pain, but the one who is most sensitive is the mother, because the husband sometimes doesn't take it seriously... it ends up seeming as if it is a she who decides, while it is the husband who decides.*
>
> *Participant 5: The first person who sees that the child is sick is mummy, and then the husband certifies that really the child is not well; we both decide to take the child to the health facility because we are together*" (FGD 03, men, household head, Mapulangune).
>
> "*When I see that the child is sick I inform my husband, and he tells me to take him to a health facility. When I inform him and he says 'for this illness you don't go to the health facility, but go to the* [traditional] *healers', then I go to where he decides*" (FGD 02, women without decision-making power, Magude village).
>
> "*Who decides* [that the child has to go to the health facility] *here at home is me.* [And if I am absent] *my wife is the one who decides*" (SSI 01, man, household head, Motaze).

In contrast to other male participants, community leaders in particular said that in case of healthcare for children, women took the decision about when and where to seek healthcare even in the presence of the husband, because women were the caretakers of the children and were more sensitive to the health status of the children. In fact, all women with decision-making power included in this study, reported deciding about when and where to seek healthcare for their children and themselves. Within this group, some participants who were also widows or single alluded that they took the decision to seek healthcare for themselves, while other participants who were living with their partners said that they first took the decision to seek healthcare, and later informed their partners.

> "*Who is in front of all this* [children health] *is my wife, because she is the one who is always close to the children, she is the one who has the right to take the child to the health facility when she sees that the child is not well*" (SSI 02, community leader, Panjane).
>
> "*Researcher: When someone has fevers, who decides that you have to take that person with fever to health facility?*
>
> *Participant 1: You alone are the one who sees that your person is sick and you take him/her to health facility.*
>
> *Participant 6: I take the child to the health facility, and when I come back I tell my husband that I took the child to the health facility.*
>
> *Participant 3: I am the one who is responsible, I am the one who sees that this child is sick.*

*Participant 9: I am the one who feels the weight of the child's illness, I only call him* [my husband] *to inform him, but when I see that the child is in a lot of pain, I take him to the health facility*" (FGD 02, women with decision-making power, Mahel).

When children experienced fever at school, the teachers took the decision to seek healthcare after evaluating whether it is simple or severe. They reported taking the child immediately to the health facility when they considered the fever to be severe.

"*When the child says 'Mr. teacher I'm not well', I ask if is a simple headache or serious pain, and if it is a serious problem, I communicate to the* [school] *direction (...), then I fill in the guide, I request the auxiliary personnel to take the child to the health facility. If the auxiliary is absent, I take the child to the health facility*" (SSI 05, teacher, Panjane).

With regards to the timing to seek healthcare, several participants living near a health facility or a CHWs, regardless of whether their place of residence was rural or urban and of their administrative post within the Magude district, reported seeking healthcare immediately after the onset of the fever, because they were afraid the illness could worsen or even cause death. This was particularly the case for caretakers who had the power to immediately decide on when and where to seek care for children, mainly women with decision-making power.

"*Hiiii..., if you take days, you are already accumulating the disease, and it can kill the child, it has to be soon, the child is taken to the health facility immediately because the disease can be serious. The child may look better and then suddenly he can get worse at night (...)*" (ISS 04, Community Leader, Mahel).

"Participant 7: *Sometimes when I see that it's serious, I don't take too long, I take him to the health facility, to get care quickly. If I want to wait and see if it gets worse, the person could lose his life. I take less than a half of an hour to get to the health facility*"

*Participant: 3: When I see that the child is sick... I don't take 1 hour, I take the child immediately* to the *health facility.*

*Participant 2: When you live close to the health facility it doesn't take much time, only for those who live far away, they take time*" (FGD 02, women with decision-making power, Panjane).

Participants living far from the CHWs or health facility said that sometimes they could not seek healthcare immediately when the fever symptoms started at afternoon or at night due to the lack of transport, lack of money for transport, or when they knew that the health facility was already closed, as stated below by women without decision-making power within the family and by male household heads. However, others delayed seeking healthcare because they first evaluated the fever symptoms before deciding to seek healthcare, regardless of the distance to the health facility.

"*I would say that it depends on the time when the child starts getting sick, if it starts in the afternoon, I have no way to take him to the health facility, because there is a lack of chapas* [local public transport]*. I wait until early in the morning and I take him to the health facility. If he starts getting sick early in the morning, I take him immediately (…)*" (FGD 02, women without decision-making power, Magude village).

"*(…) Our health facility here in Mapulanguene only functions during the day, at night it is closed. There are cases in which the fever starts at night (…), if it starts at 6 P.M., I take him to the health facility, but if it starts at 7 p.m. by then I can no longer take him to the health facility because another health facility is far from here*" (FGD 03, men, household heads, Mapulanguene).

"*I don't take him* [the child] *immediately, first, I observe, I wait until tomorrow, to see how he slept during the night. If I don't see improvement, I take him to the health facility*" (SSI 05, woman with decision-making power, Magude village).

Both health professionals and CHWs unanimously mentioned that some patients sought healthcare immediately, while others delayed care seeking between one and three days after experiencing fever symptoms. They perceived that care was sought immediately when it was children who had a cough, high body temperature and/or a headache, and they delayed it when adults were ill. With regards to differences in timing to seek healthcare across genders and community groups, health professionals perceived that women with young children were quicker to seek healthcare than the youth and adult men and women, because they were concerned with their children when they were sick. But adult patients, particularly men and the youth, often delayed seeking healthcare for themselves and they may first use home remedies. Nonetheless, health professionals pointed that women, in particular, sought care at the health facility earlier than men.

"*(…) There are those* [patients] *who when they feel fevers they come at that moment, there are those who may feel it today and it may pass two days, the third day, then they come for consultations, it is not the whole population*" (SSI 01, CHW, Magude village).

"*Many of the times they are women, or because they come to accompany a child, in this case children are the ones who come the most to the consultations, accompanied by their mothers or aunts, grandmothers, but also women themselves when they are feeling bad* [fever], *they come to the health facility, unlike men and young people who come when they are very bad, after a day, two or three days, mostly after taking home remedies*" (SSI 02, health professional, Motaze).

**Healthcare-seeking at health facilities.**  Several participants of the different community groups included in this study, regardless of their place of residence, said that they often considered health facilities as their first choice whenever they perceived illnesses, such as diarrhoea, body pain, headache, fever, sores, thrush, malaria, cholera, body warming, stomach-ache, vomiting, weakness, HIV/AIDS, tuberculosis, arterial hypertension, uterine pain, swollen feet and paralysis. Participants chose the health facility as their first choice when they considered it was near their homes and when they perceived that CHWs were not able to test and treat some relevant diseases (such as HIV/AIDS and tuberculosis). Moreover, participants added that they preferred to go to the health facility to access the diagnosis of several illnesses and know their causes, access medication and treatment. Participants also mentioned that they often followed the community health campaign that recommended that everyone goes to the health facility whenever they are sick, and they perceived that the medicine from the health facility successfully treated diseases.

"*The health facility helps us, it tests to see what sickness you are suffering from, then it gives you pills (…) health facility helps because if the person lacks water, blood, it gives water* [serum], *it gives blood and you are cured*" (FGD 02, women with decision-making power, Panjane).

With regards to fever cases, several participants reported preferring to seek healthcare for fever at the health facilities, regardless of the severity of the fever (simple or severe fever). But some only sought care at health facilities after CHWs failed to treat the fever or when they perceived fever as a "*serious fever*".

"*Researcher: When you or the children feel fever, where do you go first to treat the fever?*

*Participant: To the health facility, here in Mahel, there is no other place. I treat it at the health facility, whether it is for headaches or stomach aches, because I know that health professionals will give me pills to treat it (…)*" (SSI 04, community leader, Mahel).

"*I can see in the child if the fever she/he has is serious or not, if I see that this fever is serious, I go to the health facility. (...) When my child has fever, I go to the CHW. But if the CHW doesn't manage it, he gives pills and I see no improvement, then I take* [the child] *to the health facility (...)*" (SSI 03, woman without decision-making power, Panjane).

The preference for health facilities was often related to the perception that the health facilities could manage all types of fevers. Moreover, participants associated fever to malaria, which some believed should be treated at health facilities mainly, while others perceived that fever could even cause death when it is not treated.

"*Participant 2: Only the health facility can manage to treat all fevers because it is related to malaria, (…), malaria is treated in the health facility. Fever is related to malaria, and only the health facility can treat it.*

*Participant 5: The health facility is where we grew up until we reached this age, when you're sick you have to go to the health facility, there's nothing else*" (FGD 02, men, household heads, Motaze).

"*It's because we want help, because that mututumelo* [fever] *can kill the child, or me for that matter, because if I don't go to the health facility it will get worse and I might die. We go to the health facility to get pills or be cured*" (FGD 04, women without decision-making power, Mahel).

Health professionals confirmed that most members of the Magude district mostly sought care at health facilities to treat fever cases, especially in children because they mostly associated it to malaria.

"*Well, here in the Magude district, fever has been one of the main causes that lead patients to seek treatment in the health facility (...), especially in children, when the child gets warmness, the patient or parents and caregivers immediately rush to seek healthcare*" (SSI 03, health professional, Magude village).

**Healthcare-seeking at Community Health Workers.** Participants often sought CHWs mainly when they perceived the following symptoms and illnesses in both adults and children: warm body, body pain, fever, headache, loss of appetite, stomach-ache, vomiting, diarrhoea and malaria. The use of CHWs was linked to the perception that CHWs were allocated to help the community, were close to their home in contrast to the health facility being far, and were able to treat fever and malaria.

"*What makes us go to the CHW is that she was allocated here so that when we feel a headache or pain in the body, because it is close by, we go there, because the health facility is far away. So, we go there to test, because CHW also does malaria tests. If CHW detects malaria, she gives you malaria tablets, if it's fever, she gives you fever tablets to cure the disease*" (SSI 01, man, household head, Moatze).

Regarding fever in particular, participants often sought CHWs as the first choice in the beginning of the illness and when they perceived that the type of fever they were suffering from "*is not serious*" and could be handled by the CHWs. These participants perceived that CHWs could treat simple fevers, while severe fevers were treated at the health facilities.

"*When the illness is starting, I go to the CHW because I know that* CHW *is someone who heals, gives first aid, as soon as the disease starts I go there, when he gives first aid I get better. Only when I see that it is not getting better, then I look for help in the health facility*" (SSI 02, man, household head, Magude village).

"*(…). Here in the community you have help, but only if the fever is not serious, (…) Here we ask first the CHW to give us first aid when it's a headache... but the important thing is to go to the health facility*" (FGD 03, women without decision-making power, Motaze).

The use of CHWs as first choice was mostly reported among participants living in remote areas, where the health facilities were far from their homes. Even though the general preference for treatment of fever was health facilities across genders and among women both with and without decision-making power. For many participants CHWs were their first feasible and available choice of healthcare due to their distance to health facilities. Moreover, participants reported receiving advice from health professionals to use CHWs. However, participants claimed that CHWs were limited in terms of diagnostic capacity and access to medicines, and they argued that CHWs were not able to deliver a complete diagnosis, determine the type of illness and provide a complete treatment.

"*Participant 5: We have that first-aider* [CHW] *who can't treat you with the medicine he has; and he accompanies you to Motaze health facility. But in Motaze, when you are diagnosed lack of water in your body, it means that the malaria has worsened, they call the ambulance and you are transferred to Magude* [referral main health centre]*.*

*Participant 3: (…) I say it's not the same thing, because that one from the community* [CHW] *won't know where the illness is going, if I go to the health facility they will observe me, if it's serious, they will quickly refer me to Magude* [referral main health centre]*, because they will observe me, if it's serious or not, if it's not serious they will treat me locally*" (FGD 03, women without decision-making power, Motaze).

In fact, CHWs recognized that they had limited capacity regarding the diagnosis and treatment of several illnesses. They mostly reported treating fever when they have the appropriate medications. They said that they often referred patients to the nearest health facility for further diagnosis or treatment when they could not treat the perceived illness.

**Healthcare-seeking at pharmacies and informal drug salespersons.** Participants also bought pills from private pharmacies and in informal drug sellers to treat fever related symptoms. Three participants reported self-medicating using medicines obtained from their neighbours, private pharmacies, or informal drug sellers when they felt a headache or were coughing. This practice was mostly common in the earliest stages of the illness, when they perceived that the local health facility was already closed and they lacked resources to seek care in other health facilities.

"*Let's say... for example, I'm starting now not to feel well, it's now fourteen o'clock, so they've already closed the health facility here in the community, I'm not talking about Magude* [referral main health centre]*, because Magude still has the emergency ward. Because the health facility here in Facazissa* [name of the community] *is already closed, I can only go and ask a neighbour for pills, and if I see that the pain doesn't go away, then the next day I will go to the health facility*" (SSI 05, women without decision-making power, Magude village).

Moreover, a number of participants self-medicated using medicines from pharmacies and informal drugs sellers to treat illnesses such as malaria, body pain, back pain, feet pain, sole pain and bone pain, headache, and stomach-ache. The participant's reasons for using medicines bought at the pharmacy and from informal drug sellers was related to several perceptions: i) when they perceived an immediate illness and they thought the health facility was far and costly, ii) when they knew of or suspected of the potential shortage of tablets at the local health facilities' pharmacy, or iii) when they considered it more costly to seek care at a health facility for an illness such as headache than to buy medicines from the local vendors.

"*It has happene* [that people buy pills from the street people] *when a sudden illness arises and the person thinks that he/she cannot make it to Motaze* [health facility] *(…). It happens that the person gets malaria from here to here, the person may have money there and buy it*" (SSI 04, women with decision-making power, Motaze).

"*What makes us buy pills from the informal vendors is that we think we have a headache, but we don't have 200.00Mt* [USD 3.3 for round trip transport fare] *to go to Panjane health facility. Now, if the informal vendors come and you have a*

*headache, you can buy pills with 100.00Mt* [USD 1.65] *(...)*" (FGD 01, elders who were caregivers of children under 15 years, Panjane).

Household heads and elders with or without a caregiving role, in particular, reported using informal drug sellers due to a lack of an adequate treatment at the health facilities in issues related to sexuality, as the following narrative highlights.

"*Participant 3: Another thing is related to us elder people, when we go to the health facility we no longer get the help we need when we are sick, but at the informal drug vendors, we can get help. We, the elderly, we used to take a virility potion, which made us sexually active (…), but now when we go to the health facility, we are even ashamed to ask about virility pills, because I don't even know how to approach this subject.*

*Participant 2: When we ask about the virility pills at the health facility, they say that we don't work anymore* [are no longer sexually active] *because we have already grown up. The nurses start laughing and making fun of us. They say: 'what do you want to do, you're already old, what do you still want to do?' But we want to, because (…) the lack of virility is a real problem (...). But the informal vendors give us pills for virility*" (FGD 02, men, household heads, Motaze).

Health professionals and CHWs confirmed that some patients self-medicated and used drugs acquired without medical prescription to treat fever, particularly when they have a cough, before seeking a CHW or a health facility. They said that patients often alleged that they thought it was simple cough, and it would go away.

**Healthcare-seeking at the traditional healers.** No participant reported seeking healthcare at the traditional healers as their first preferred option for the treatment of fever. However, some participants believed that fever could be treated at the traditional healers, especially after seeking treatment at the health facility and the health facility having failed to treat the fever, as the following participants' discourses show.

"*Yes. There are people who treat* [fever] *traditionally, they will take medicine they say it is for the cough to go away (…).* [And] *not only cough, when they see that the child has a high body temperature, they run to seek help from traditional healers*" (SSI 12, woman without decision-making power, Magude village).

"*Yes, when the child vomits until it loses its strength, I take the child to the health facility, and when I see that the child is still sick, I take her/him to the traditional healers, they will give the medicine to the child*" (FGD 02, women with decision-making power, Panjane).

Traditional healers confirmed that they only treated fever related to superstition traditionally, but they emphasized that they often performed the treatment after the health facility failed to cure it. However, one of the traditional healers claimed that he treated cough traditionally as presented in the following discourse.

"*I am the one who knows the cure, it could be cough or other disease (…) I traditionally treat* [cough]" (SSI 03, traditional healer, Magude village).

Health professionals said that although people sought health facilities for treatment, patients also went to traditional healers after the health facility consultation. They added that this practice was common when patients' sicknesses took time to heal. One of the health professionals expressed her view as follows.

"*Well, patients go to traditional doctors (…). When they receive treatment for long time, they end up opening themselves, and say: 'haa no, this fever is not normal, I have already taken pills, and it is not healing, we have to see*

*something else'. So, we advise patients to use traditional medicines, but not to mix traditional medicines with modern medicine*" (SSI 01, health professional, Panjane).

**Self-treatment at home.** Home treatment of fever was mainly recurred to as an alternative of treatment at formal healthcare services when the distance to them was seen as a barrier to reach them or when they perceived fever as not "severe illness" that needed to be treated at healthcare facilities. In a such cases, participants reported using home-made syrups to treat fever, cough and colds.

"*Today I have a cough. It started yesterday… I made syrup. … It is still not strong* [so it is not a cough that needs going to the health facility] … *Yes* [it is that cold that one is able to treat oneself at home]" (SSI 03, pregnant woman, Magude village).

Moreover, home treatment was also employed for non-febrile diseases such as wounds and pain. Participants reported using home-made remedies to treat illnesses such as colds, headache, stomach-ache, body pain, back pain, feet and sole pain, joint pain, bone pain, asthma, and wounds. They reported using self-treatment of some illnesses at home because they perceived that these illnesses were not serious enough to require a health facility, when the illness stared suddenly at night and they lived far from the health facility, or after sensing that the prescribed pills were not immediately treating the illness.

"*Participant 4: What leads us to use home remedies is that sometimes the illness can attack me from here to here and our health facility is far away, sometimes I get up late at night without a car and I see that the health facility is far away, I won't be able to get there* [by foot]. *At that moment I look for traditional medicine*" (FGD 01, women without decision-making power, Panjane).

"*First, we take him* [child] *to the health facility, when we see that after giving him pills he doesn't improve, when he continues vomiting, we look for other plants* [home-made syrups] *to treat the vomiting, and only after that he takes pills*" (FGD 02, women with decision-making power, Panjane).

Aligned with the above discourse, health professionals and CHWs also said that some patients self-medicated, using home remedies such as home syrups to treat fever, particularly when they have a cough, before seeking a CHW or a health facility.

**Perceived facilitators to seek care for fever within the available formal health services**

All study participants, regardless of their place of residence and across all community groups, perceived that the facilitators to seek care for fever within the formal healthcare services were the availability of CHWs and health facilities near their homes, access to health services at any time, availability of medicines at the health facility pharmacies at almost no cost, and the belief that the medicines offered at the health facility treats fever.

"*I get support in the health facility, when I arrive there* [health facility], *they treat me quickly, when I am sick they do not charge money (...), they observe my illness and give me the pills that cost 1.00Mt or 2.00Mt*" (FGD 04, men, household heads, Mahel).

"*What makes it easy for us, is that the CHW is very close to our homes and when we call him he doesn't deny coming to our homes to cure us. Sometimes we call... he already knows that when you call, it's because you can't reach him. When we can, we go to him, because it's closer than the health facility. Even if you call at night, he comes to meet you, because he has a bicycle that they gave him for that*" (FGD 01, women without decision-making power, Panjane).

Moreover, participants said that they were happy about the procedure used to diagnose fever at the health facility, which included testing and disclosure of the results, and they said that they were well attended by both CHWs and health professionals at health facilities. As a matter of fact, both health professionals and CHWs reported their full availability to attend patients at any time. CHWs, in particular, said that they attended patients anywhere and anytime, while health professionals affirmed that patients did not take a long time to be attended at the health facility.

> "*Even if I'm working in the field* [agriculture]*, I attend the patient who is in need. Even today, I attended three patients while I was in the field*" (SSI 04, CHW, Panjane).

> "*Here the patients don't wait so long, when a patient arrives, even if they arrive while I am already at home, I go to the health facility and attend the patient (...)*" (SSI 04, health professional, Mapulanguene).

Participants who were living far from the health facility or from a CHWs added that other facilitators were the availability of public transport, having transport money, or a neighbour with a car who could help when someone contracted an illness at night. Furthermore, having someone to take care of the house and children when seeking for healthcare enabled participants to seek healthcare at the health facilities. Several participants said that they had someone to take care of the children or the house. The caretaker could be a family member, such as husbands, grandparents, aunts and sisters, or neighbours.

> "*When* [the illness] *gets worse at night, we turn to the neighbours, who we know have a car, if they can, they will accompany us, if they can't we walk, because we are far from the health facility*" (SSI 03, woman without decision-making power, Panjane).

> "*Participant 8: It can be myself, or mummy with the child, but when the mother takes the child to the health facility, I stay at home to take care of the house.*

> *Participant 2: When I take a sick child to health facility, my parents or neighbours take care of my house*" (FGD 01, men, household heads, Magude village).

Participants also said that they were motivated to seek healthcare for fever within the available formal health services because the Government, CHWs and health professionals always sensitised them to reach out to health facilities or CHWs whenever they contracted fever.

> *We go to the health facility because it is a norm imposed by the government. Yes, they said we have to go to the health facility, that's where we will get help*" (SSI 04, elder who was not a caregiver, Magude village).

In fact, health professionals and CHWs reported performing continuous community engagement tasks and recommending communities to seek care at health facilities or at CHWs whenever they experienced any illness. Health professionals, in particular, said that they have a collaboration framework with traditional healers which consists in sensitising people to seek healthcare at the health facility before reaching out to traditional healers. This collaboration enables traditional healers to refer their patients to the health facilities.

> "*We have established some points of collaboration and cooperation with traditional healers, this is the reason why today, there are guias de transferência* [referral documents] *from the traditional medicine practitioners to health facilities. When traditional healers receive a patient, they refer* [the patient] *to the health facility (...). We have even periodic meetings with traditional healers*" (SSI 03, health professional, Magude village).

**Perceived barriers to seek care for fever within the available formal health services**

All participants, regardless of their place of residence, perceived several barriers when seeking healthcare for fever within the available formal health services. Living far from the health facilities or from CHWs, lack of transport and money for transport to go to the health facility, lack of access to healthcare services after working hours (7h30 to 15h30) and during weekends were reported as the main barriers.

> "*From here, it takes 3 hours to get to the nearest health facility, which is Captine. Another health facility in Panjane is 27 kilometres away. The health facility there* [Captine] *closes on Saturdays and Sundays*" (FGD 01, elders who were caregivers of children under 15 years old, Panjane).

> "*(...) We are not happy with the working hours. The health facility closes around fifteen o'clock. We really don't like the time (...). For example, if I start not feeling well now, it is now fourteen o'clock, so they have already closed here in the health facility in Facazissa* (name of local community) *(...)*" (SSI 05, women without decision-making power, Magude village).

Another barrier was a perceived lack of access to adequate service in the health facilities. Several participants perceived numerous hindrances in the health facilities when seeking healthcare for fever, including long queues and long waiting hours to access services, lack of correct diagnosis of the illness, lack of correct prescriptions of treatment and lack of the prescribed medicines in the local health facilities' pharmacies.

> "*When we arrive at the health facility, we make an appointment, they do tests to find out if the person has malaria or other diseases, if they don't find malaria, our health facility doesn't give the right tablets, the tablets we get from the health professionals are for malaria and headaches, but for stomach aches, they don't give me the right tablets, it happens that I complain of stomach aches and they give me tablets for headaches*" (FGD 02, men, household heads, Motaze).

Participants said that when they are not attended adequately they either seek care in another health facility or give up, preferring self-medication or traditional healers as alternative therapies. In addition, when the medicines were not available in the health facility, participants bought them in the private pharmacies, which implied a high cost.

> "*Sometimes you go to the health facility and after analysis they tell you to go back home, as long as you are not satisfied even other medicines you may find them useful to take, because you were not treated at the health facility. Some end up looking for traditional healers to treat the illness*" (FGD 01, men, household heads, Magude village).

Health professionals' absences and their attitudes regarding patients were also considered barriers to seek care. Several participants perceived that some health professionals were often absent from the health facilities, did not attend patients "*very well*", they did not always test the patients and when testing, not all health professionals observed confidentiality during the disclosure of the test results, and they did not give priority to patients with fever symptoms.

> "*When I arrive at the health facility, sometimes the nurse sees things that are not right, she goes outside and says "that child has disease x and y". Instead of telling me, she goes outside and tells everyone and when I leave, I look crazy... it's one of the things I condemn*" (FGD 02, women without decision-making power, Magude village).

> "*(...) When a patient has muzothotho* [fever]*, they don't give priority, that is what discourages us. When child has pain, they don't give priority in our health facility. Sometimes, they attend, and give us the pills, but sometimes, I don't know if they are angry or what, because when you tell them that you have muzothotho, they don't test you, they only give you*

*pills, and sometimes you get worse and return* [to the health facility] *(...)*" (...) (SSI 02, woman without decision-making power, Magude village).

In addition, one of the participants pointed out corruption at the health facility as one of the barriers to access adequate services quickly, while others associated lack of adequate health services with frequent reallocation of health professionals who in their perceptions were very young, lacked adequate knowledge and experience, and did not respect the patients.

"*Participant 7: The service here in Motaze health facility is not good, health professionals who come and work here in Motaze don't know anything, both in maternity or even in the medicine department (…). Once they start learning, they are exchanged, and new ones are sent here. (…). They send some girls, I don't know, so young...*

*Participant 3: Having trainees in the health facility is hurting us, because they are not grown up women, they don't have respect and they do not know how to treat people.*

*Participant 1: In short, the people* [health professionals] *who are there at the health facility have no experience, they come to learn how to work*" (FGD 02, men, household heads, Motaze).

The use of traditional healers was also pointed out as one of the barriers to timely seek care for fever within the available formal health services. Participants from different groups included in this study said that some people first sought traditional healers whenever they felt an illness, including fever. However, traditional healers reported that when people sought them for help before going to the health facility, they advised patients to seek healthcare in the health facility first before initiating traditional treatment.

"*Some people come when they have fevers, but first, I tell them to go to the health facility to be tested. Then, after that, they come here. Some people end in the health facility, they don't come here anymore. Here, they know that they will get help, because it can happen that the person has spirits, it can happen that the person has demons, so they come to know about it*" (SSI 05, traditional healer, Motaze).

## Discussion

This study analysed healthcare-seeking behaviour within the framework of the local conceptualization of illness in general terms and of fever in particular in the Magude district population. Specifically, the study identified perceptions about fever, described decision-making process and healthcare-seeking behaviour for fever cases, and identified facilitators and barriers to seek care for fever within the available formal health services.

The findings show that, in this district, febrile illness is understood to be part of the common general collective illnesses and it is mostly identified and interpreted according to local lay knowledge based on physical and behavioural signs, and on the experience of different illnesses. The study revealed a local construct of "fever", which includes but is not exclusive to a high body temperature, contrasting the biomedical concept of fever, which is based on a body axillary temperature equal or above 37,5°C [1,48]. In fact, participants pointed to other informative physical and behavioural signs that do not require a measurement of the body temperature, which suggest that an adult or a child has a fever, and are used as indicators for fever disease in the given context where the majority of the Magude population does not possess a thermometer at home. Thus, fever was perceived as an illness often accompanied by a set of signs and symptoms, including body pain, high body temperature, coughing, vomiting, red eyes, shaking, stomach-ache, feeling cold, and headache. This implies participants perceived they had the capacity to detect fever based on their observations and experience. Similar results were also documented in Uganda, where communities also associated fever to multiple temperature elevating disease processes [49].

Despite the ability of the participants to identify fever, the signs used have implications for the healthcare-seeking timescale, because if the unusually high body temperature is not detected quickly by caregivers, which is one of the first signs of infection, and caregivers are mainly alerted to the existence of fever by other signs that usually appear in later stages of febrile illnesses, such as vomiting, diarrhoea, loss of appetite, or lethargy, this may imply a delay in healthcare-seeking. As the results of this study show, fever was locally categorized as simple fever or severe fever that could require formal health services or other non-formal local sources of treatment. This finding corroborates with previous findings in Mozambique [37], Tanzania [50,51], and Uganda [49], which highlighted that people classified fever between normal expected illness and serious illness.

Although previous studies in the study setting [28] and elsewhere [52–55] showed that participants associated fever to malaria, and predicted that this could be the main trigger for healthcare-seeking in the available formal health services, the findings of this study show that not all participants associated all types of fever to malaria, despite correct identification of malaria symptoms (including fever) and mosquito bites as the cause of malaria, because participants were aware that fever could have other causes other than malaria infection, as it was also documented in Uganda [49]. Participants of this study mostly associated more severe forms of fever to malaria [28,49] and some participants perceived that not all types of fever severities required care at formal health services, as other studies showed in Tanzania [50,51] and Uganda [49]. Therefore, for even mild fevers to become one of the main triggers for healthcare-seeking within the available formal health services there is a need to increase community awareness, focusing on different perceived fever categories and their association to malaria [28,56,57] and other diseases like pneumonia [58], as health professionals suggested in this study, and to make formal health services available and accessible to the whole population. The categorization of fever and its association or not to malaria derives from individual and collective lay conceptualization, experience with the illness and interaction with health professionals. For example, the fact that the Magude population has been experiencing different symptoms of fever, that can be diagnosed as malaria or not by health professionals, creates an opportunity to local population to learn and construct their own local aetiology, from which they sense, categorize and take decisions about fever management.

In fact, the local perception of fever has implications in timing healthcare-seeking for fever in formal healthcare facilities. The findings of this study reveal that the decision on when to seek healthcare within the available formal health services was also shaped within the family level by the evaluation of the fever signs and the perceived availability of the formal services. Participants reported seeking healthcare immediately after perceiving fever signs and when they perceived that they could access health services, while others delayed healthcare seeking because they first evaluated the severity of the fever specially if they perceived that the formal health services were far from their residences or were already closed. According to CHWs and health professionals, patients only sought healthcare immediately for fever cases among children, as it was also documented in Kenya [59] and Malawi [33]. But they often took two or three days to seek care when adults experienced the same illness, as it was early reported in Ghana [60], Zambia [61] and Ethiopia [52]. Other studies about healthcare-seeking among children with fever in Ethiopia showed that communities did not seek care for fever because they perceived it as mild disease [62] and, in Tanzania, mother delayed to seek care for malaria among their children because they believed they were dealing with ordinary fever [63]. Other studies among children and adults in other sub-Saharan countries highlighted that patients rarely sought healthcare for fever cases at the formal health services within 24 hours after onset of a fever or a cough [34,64–71]. The contrasting finding that some participants living close to health services sought care for children immediately upon the presentation of fever in this study may be related to the fact that the community under study has been part of a malaria elimination campaign since 2015, and the local population was exposed to community awareness campaign against malaria, which stressed the importance of seeking formal healthcare services after the onset of the fever.

Healthcare-seeking also depends on who decides when or where to seek care. In this study the choice of the source of fever care was influenced by the individual or collective perceptions of fever at home or at the family level. As Kleinman

[72] highlighted, illness is first defined at home and healthcare activities are initiated there, at the family level, upon individual or collective perceptions and beliefs about the illnesses, and it is within the family that the decision to seek healthcare in the health facilities or traditional healers is initiated. Results of this study show that the decision about where to seek healthcare for fever cases laid mostly within the family, varying between the male household head, the woman within a caretaker role, or a collective decision including both wife and husband or other members of the family, as other studies also documented in India [73], Senegal [74] and other Sub-Sharan African countries [75].

Collective decision-making and the lack of decision-making powers held by women have been documented as factors contributing to the delay to seek healthcare [50,74,75]. Conversely, when women have power to decide, they sought appropriate healthcare for fever cases early on, particularly for children [64,76–78]. Our study also revealed that women with decision-making power took the decision to immediately seek healthcare for fever cases, particularly for children. Nonetheless, some women with decision-making power reported seeking care the day after the disease started (not immediately, as others reported) because they first evaluated the "seriousness" of the illness. In general, women's role in the decision-making process for healthcare for children seems to be related to their common culturally and socially assigned activities of childcare [79,80] and their marital status, since some women had decision-making powers because they are considered responsible for childcare, while others had such power because they were singles or widows as showed in this study. Although this result highlights a potential opportunity to increase healthcare-seeking among children and women through further women empowerment in healthcare, there is a need for further studies to investigate how married women negotiate and use their decision-making power, and how such decision-making power influences healthcare-seeking within the formal healthcare system. Upon finding that community leaders, in contrast to other men, considered their wives to be the main decision-makers about healthcare seeking, this study's findings suggest that community leaders should incentivise other husbands to extend decision-making powers to women for the healthcare-seeking of children and for themselves as a strategy to mitigate behavioural-based delays. This echoes the proposal by Ewing et al. [81] of involving both women or those with a caring role and all the key persons involved in healthcare-seeking decision-making processes in order to empower and promote the ability of women bearing the caring role to seek timely care for themselves and for children or other family members they care for.

The study's results reveal that the participants' healthcare-seeking behaviour was an interactive process within and between the available non-formal and formal health services, depending on the way fever was locally perceived and experienced. Some participants self-medicated first, then sough care at CHWs or health facilities and later sough traditional healers, while others first sought care at CHWs or health facilities and later self-medicated or sought traditional healers. The practice of interactive processes between non-formal and formal health services was also documented in Tanzania, where it was shown that healthcare-seeking was reiterative, cyclical and dynamic [51]. As the results of this study also show, the categorization of fever into simple and severe fever allowed families to assess the opportunity cost of seeking healthcare at health facilities in light of the severity of the fever and other factors related to the feasibility and the burden of seeking care at the health facility such as the distance to the health facility, closing times, and availability of transport. Thus, despite reporting a preference to seek care at health facilities, this was not always feasible or realistic and participants used multiple sources to treat fever, including both non-formal health services, such as self-medication and traditional healers, and formal health services, like CHWs and health facilities. Self-treatment based on using home remedies or on medicines bought from private pharmacies or informal drug sellers was used when participants perceived suffering from a simple fever or a cough, while CHWs or health facilities were sought when they perceived severe fever or after self-treatment failed to cure the fever, or when they associated fever to malaria. In addition, traditional healers were sought when fever was perceived as an illness related to superstition, before or after experiencing the failure of fever treatment at the health facilities, as it was also documented in Tanzania [50,51,55], Uganda [49], and Burkina Faso [82]. The use of multiple sources of care for fever was also observed in other previous studies in Sub-Sharan African [68,71,76,83–85] where biomedical concept relating to fever aetiology also co-live with traditional concepts of illnesses along with spiritual origins, as is the case in Mozambique.

Several factors influence the use or non-use of formal healthcare services to treat fever cases. The findings of this study show that despite the availability and use of multiple sources to treat fever cases, participants preferred health facilities and CHWs for fever treatment, due to the following perceived facilitators: the health facilities were perceived as capable to diagnose and manage all types of fevers, fever symptoms were believed to be often associated to malaria, participants had the perception that malaria could only be treated at the health facilities or CHWs, and also due to the perception that fever could lead to death if left untreated. Other factors facilitating the immediate seeking of care were the accessibility of those services at any time (which was reported for CHWs), the opportunity to access medicines at almost no cost through the formal health services, and the belief that the medical services offered by the health facilities treat fevers. The finding that proximity of formal health services and availability of health professionals to deliver healthcare increase the probability of healthcare-seeking is corroborated by other studies in Mozambique [86], Ghana [84], Nigeria [68], Burkina Faso [82,87], Ethiopia [62], and other sub-Sharan countries [52,66]. This study also shows that the availability of public transport, having money to pay for transport or having a neighbour with a car who could help when someone contracted an illness at night were also considered as additional facilitators among participants living far from health facilities. Access to economic resources and availability of public transport have also been reported as facilitators to seek care in Sub-Saharan Africa countries [75]. In addition, community engagement and awareness campaigns played an important role since most participants were aware that they had to seek formal health services whenever they experienced an illness, including fever. Health education through awareness campaign on fever in the community was also considered an important factor to uptake healthcare-seeking within the formal health services in Nigeria [88,89] and Burkina Faso [82] and elsewhere [90].

However, access to formal health services for fever cases was constrained by a number of structural barriers faced by the participants, including the distance to health facilities, lack of transport and lack of money for transport to health facilities, and lack of access of health services after working hours and during weekends, as other studies showed in Tanzania [70] and Uganda [49,71], where the ratio of health facilities and health care workers to population is also sub-optimal. At the health facility level, participants also faced several barriers, such as long queues and long waiting hours to access services, inaccurate diagnosis, incorrect prescriptions and lack of the prescribed medicines in the local health facilities' pharmacies, as it was previously documented in Mozambique [36,86], Uganda [49,71], Tanzania [51], and other sub-Saharan countries [50,52,66]. Additionally, participants also presented other barriers related to service delivery, such as the lack of priority of fever cases, inadequate service delivery by health professionals which included not testing fever cases using malaria tests (which the population expected when they had a fever), not respecting the patient confidentiality in the disclosure of the test results, health professionals using their working hours for personal affairs, and corruption. The perception that not all symptoms associated to fever required formal healthcare treatment (for example, cough or stomach-ache were believed to be manageable at home), the use of self-treatment at home, and the perceived association of some fevers with witchcraft specially after treatment having failed at health facilities also hindered or delayed healthcare-seeking for fever cases, as observed in other studies [49,84]. Barriers related to perceptions of fever and use of self-treatment were also documented in previous studies in Mozambique [86], Tanzania [50,51,55], Uganda [49], Nigeria [68], and Burkina Faso [82]. In addition, the use of traditional healers for fever cases has been found to be practiced in some sub-Saharan countries [52,61,76,82,91] although they were not reported as a first choice for fever treatment in the current study. The collaboration between traditional healers and the formal health system in Mozambique by which traditional healers are expected to refer patients with malaria symptoms to the formal health system may be considered one of the reasons why the participants of this study did not consider traditional healers as the first option for fever cases. On the other hand, the communities that reported a preference for traditional healers in other studies may have a more intense practice of traditional beliefs and religions than the community participating in the current study in Mozambique, which is mostly Christian. For example, the study that reported the highest proportion of participants seeking care at traditional healers as a first option (18.5%) took place in an Ethiopian community where the majority of the population practiced traditional beliefs (48%), followed by a 39% of Ethiopian Orthodox Christians and 8% of Muslims [52].

In light of the potential growth of resistance to antimalarials in the coming years, it will be important to avoid losing persistent malaria cases out of the healthcare system and into traditional healers' remedies owed to the belief that persistent fevers may have a spiritual cause. Therefore, the concept that recurrent or persistent fevers can have a biomedical cause and also require medical treatment at the health facilities should be incorporated in the framework of the collaboration between traditional healers and health professionals to ensure that traditional healers incentivise community members to seek healthcare for all fevers, both spontaneous and persistent fevers, at formal health services. Hence, the collaboration with traditional healers as a means to prompt access to effective treatment should be strengthened as such collaboration has proven to improve healthcare in Malawi [14], Tanzania [92] and Mauritania [93,94]. This collaboration should also be extended to informal drug sellers and private pharmacies to ensure that all patients are referred to formal healthcare. Given the large proportion of febrile children obtaining antimalarial drugs via private providers, various countries in sub-Saharan Africa have considered introducing malaria diagnostics at pharmacies [95]. Pilot projects introducing malaria diagnostic tests in private medicine retail outlets have proven feasible [96]. In the case of Magude district, malaria diagnostic test should be offered without additional cost, given the poor socioeconomic status of the rural population.

Despite the barriers outlined, when health facilities were near study participants and well-resourced to provide quality care, participants in this study reported going to a health facility upon the presentation of fever and a general preference for health facilities. This suggests that the evaluation of opportunity costs of seeking care at a health facility and the self-negotiation between sources of care (and associated delays or use of alternative sources of care) mostly come into play when health facilities are scarce and/or scarcely resourced. Previous studies in Mozambique also noted that structural barriers had a larger negative impact on healthcare seeking than individual barriers [86]. In a recent worldwide analysis, distance to healthcare was demonstrated to be negatively associated to care seeking worldwide [97]. Thus, the expansion and improvement of the health system and its infrastructural accessibility should be guaranteed if healthcare seeking behaviour and health outcomes are to be improved. Moreover, improving the accessibility of high-quality healthcare at health facilities requires multidimensional structural efforts, mainly: the expansion of the health system especially looking into building and resourcing more health facilities in the currently underserved rural areas for all the population to access high quality care at health facilities within a reasonable distance; the training and employment of sufficient health workforce to increase the current ratio of 0,65 health professionals per 1,000 population in Mozambique [98] to at least 4,45 health professionals per 1,000 population as the minimum indicated by the WHO [99]; improving the quality of services provided; the monitoring and prevention of stock outs; the improvement of the infrastructure (mainly road infrastructure) and the availability of public transport to alleviate the current dislocation obstacles and permit the timely distribution of medical products; and the sustainable financing of the health system and strengthening initiatives to enable factualizing the listed requirements. Previous studies in the neighbouring Malawi [33] and Madagascar [30] also advocate for closing the distance gap through improvements in transportation means and structures and/or the expansion of clinics into rural areas. Other studies in the region have proposed further structural improvements including the improvement of the quality of care through the appropriate staffing of the healthcare system [30,100,101] and the availability of medical supplies [14,31,62,100,101], and the general infrastructural improvements necessary for universal health care [100].

## Strengths and limitations

The strength of this study lies on the inclusion of participants from all geographical areas within the district of Magude, ensuring a diversity of views about the study object. Moreover, the study included different social categories of the population (sex, gender, social status and power relations), and a diversity of health practitioners that lead with illnesses in the community, enabling the understanding of healthcare-seeking behaviours from multiple perspectives within the community. The diversity of views from these different participants enabled triangulating results from both SSI and FGDs, which provided access to individual and collective perceptions and experiences regarding fever cases. The generated results enabled accurate and culturally grounded understanding of the complexity of healthcare-seeking behaviour about fever

cases, which laid beyond numbers. Nonetheless, the study results are limited to the study setting and methods used. The participants' selection criteria can be subject to sample bias because the study did not include all caregivers and potential patients of febrile diseases to assess actual treatment seeking practices prospectively. The study may also be affected by recall or desirability bias since the participants' reports of health-seeking behaviour about fever episodes were retrospective, which can bring about problems in accurately recalling details about timing and sources chosen and used to treat fever cases. Moreover, the cultural plurality of Mozambique also means that findings of this study cannot be generalized to all other settings or regions of the country. The study findings must be understood to belong in the context of a patrilinear rural community in the Maputo province neighbouring Gaza province (in the south of Mozambique and close to the capital city of Maputo), and different results may have been observed in matrilineal communities further away from the capital in the north of the country.

## Conclusion

This study concluded that healthcare-seeking for fever cases in Magude district was an interactive and complex process within and between the available non-formal and formal health services, and it was influenced by multiple factors, including structural, community, family and individual levels. Lack of enough availability of formal health services and perceived lack of quality of care within the available health services, including lack of drugs, hindered participants to seek care in formal health services to treat febrile disease. Moreover, individual and collective perceptions of fever emerged as important factors influencing timing and seeking of appropriate care for fever cases within the available formal health services in Magude district. These findings can be used to improve malaria control and elimination strategy by addressing simultaneously structural, community and individual factors hindering appropriate seeking of fever, as well as tailoring community health education about fever and its treatment within the framework of malaria elimination in Mozambique. It is recommended to address the identified structural, community, family and individual factors hindering healthcare-seeking within the available formal health services, need for the expansion and improvement of the local formal health services, in order to provide better health service delivery and better accessibility of health services. Moreover, the study recommends the strengthening and extension of community engagement campaigns involving local alternative sources of care (traditional healers, formal and informal drug vendors) in order to channel fever cases of any severity into the formal healthcare system.

## Supporting information

**S1 Appendix. Data collection tools-English version.** Semi-structured interviews (SSIs) guides and focus group discussions (FGD) guides used to collect qualitative data, translated from Portuguese into English.
(DOCX)

**S2 Appendix. Data collection tools-Portuguese-version.** Semi-structured interviews (SSIs) guides and focus group discussions (FGD) guides used to collect qualitative data, Portuguese version.
(DOCX)

**S3 Checklist. COREQ (Consolidated Criteria for Reporting Qualitative studies) checklist.**
(DOCX)

## Acknowledgments

We thank all study participants for offering their time and patience to speak to us and for being open about this personal matter. We extend our thanks to the Magude community in general for welcoming and engaging in the malaria elimination project of which this study is part of and to Magude's Community Health Workers for attempting to close the healthcare

gap. We are grateful to Humberto Munguambe, the head of CISM's Magude office during the study implementation, and also to the drivers who made this study logistically possible.

## Author contributions

**Conceptualization:** Beatriz Galatas, Caterina Guinovart, Francisco Saúte, Pedro Aide, Khátia Munguambe.

**Data curation:** Carlos Eduardo Cuinhane, Júlia Montaña Lopez, Hoticha Nhantumbo, Helder Djive, Ilda Murato, Khátia Munguambe.

**Formal analysis:** Carlos Eduardo Cuinhane, Julia Montaña Lopez, Neusa Torres.

**Funding acquisition:** Beatriz Galatas, Caterina Guinovart, Francisco Saúte.

**Investigation:** Julia Montaña Lopez, Hoticha Nhantumbo, Helder Djive, Ilda Murato.

**Methodology:** Julia Montaña Lopez, Beatriz Galatas, Caterina Guinovart, Pedro Aide, Khátia Munguambe.

**Project administration:** Julia Montaña Lopez.

**Resources:** Beatriz Galatas, Caterina Guinovart.

**Supervision:** Julia Montaña Lopez, Hoticha Nhantumbo, Helder Djive, Ilda Murato, Francisco Saúte, Khátia Munguambe.

**Validation:** Julia Montaña Lopez, Beatriz Galatas, Francisco Saúte, Pedro Aide, Neusa Torres, Khátia Munguambe.

**Writing – original draft:** Carlos Eduardo Cuinhane.

**Writing – review & editing:** Julia Montaña Lopez, Beatriz Galatas, Caterina Guinovart, Francisco Saúte, Pedro Aide, Neusa Torres, Khátia Munguambe.

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
