## [Decision Letter · Decision Letter 0]

3 Jun 2024

PONE-D-24-05377Understanding healthcare-seeking behaviour of fever cases in Magude district, southern Mozambique: a qualitative studyPLOS ONE

Dear Dr. Cuinhane,

Thank you for submitting your manuscript to PLOS ONE. After careful consideration, we feel that it has merit but does not fully meet PLOS ONE’s publication criteria as it currently stands. Therefore, we invite you to submit a revised version of the manuscript that addresses the points raised during the review process.<please by="" manuscript="" revised="" submit="" your="">plosone@plos.org. Please include the following items when submitting your revised manuscript:</please>

We look forward to receiving your revised manuscript.

Kind regards,

Khin Thet Wai, MBBS, MPH, MA

Academic Editor

PLOS ONE

Reviewers' comments:

Reviewer's Responses to Questions

**Comments to the Author**

1. Is the manuscript technically sound, and do the data support the conclusions?

Reviewer #1: Yes

Reviewer #2: Partly

2. Has the statistical analysis been performed appropriately and rigorously? 

Reviewer #1: N/A

Reviewer #2: N/A

3. Have the authors made all data underlying the findings in their manuscript fully available?

Reviewer #1: Yes

Reviewer #2: Yes

4. Is the manuscript presented in an intelligible fashion and written in standard English?

Reviewer #1: Yes

Reviewer #2: Yes

5. Review Comments to the Author

Reviewer #1: INTRODUCTION: Well researched and offers a comprehensive overview of similar studies on health seeking in other African contexts.

METHODS: Check use of commas and full stops when presenting numbers e.g. '63.691 inhabitants and 14.583 households' I would write with a comma, not a full stop. Similarly '2,7km (IQR 1,4-7,9 km)' I would write with a full stop, not a comma. This is perhaps a country-specific preference and should be for the editor to make final decision on.

Some abbreviations used that do not have full description e.g. IQR, NMCP, SSI etc.

Line 248 - Confronting who?

RESULTS:

Lines 418 - 436: Seems like a lot of important information is included here about different decision-making categories, but is a bit confusing to understand due to long sentences. Can it perhaps be broken down further, or use numbers to signify the different categories?

Line 458: Is women being more 'sensitive' the right word here? Perhaps more aware, or observant, or it needs to be further explained.

Line 549: What kind of health facility? e.g. health centre, clinic, public, private - perhaps more details about the health facilities can be added in the methods. Additionally, perhaps a few lines on the nature of the health system may help the reader better understand the overall health landscape in Mozambique e.g. are the facilities mainly government funded, or privately run? Are there user fees? Insurance schemes?

Line 782: Do you need to define what a 'formal' healthcare setting is here? If this includes CHWs, perhaps 'service' is better than 'setting'?

DISCUSSION: Line 953 and 963: The authors could also reference: Virhia J. Contextualising health seeking behaviours for febrile illness: Lived experiences of farmers in northern Tanzania. Health Place. 2022 Jan;73:102710. doi: 10.1016/j.healthplace.2021.102710. Epub 2021 Nov 19. PMID: 34801785 here also in addition to reference 33.

Overall, this is a well thought out and implemented research study. The findings are informative and will contribute to the wider literature on health seeking across various contexts. The only key thing I would suggest to add is perhaps providing a diagram or table or description of the overall health landscape in Mozambique identifying who and what the key health services are, whether they are formal/informal/public/private etc. There are many different services and actors mentioned in the study so it may just be useful to have some way to summarise them at the beginning of the results section. See Davis et al, 2021: “If You Do Not Take the Medicine and Complete the Dose…It Could Cause You More Trouble”: Bringing Awareness, Local Knowledge and Experience into Antimicrobial Stewardship in Tanzania - Table 2 (animal health) for a suggestion on how this could be done.

Please check minor spelling errors throughout.

REFERENCES: Reference 33 should be spelled 'Virhia' not 'Verhia'

Reviewer #2: Thank you for the change to review this paper. The study’s findings have implication for Mozambique’s malaria elimination fight. However, the manuscript will benefit from major revisions. Find below my comments:

Title & Abstract

1.Delete “Understanding” from the title.

2.The Abstract does not present the research problem necessitating the study. “Qualitative approach” is not a study design. The authors should state the exact study design the used for the study. Data collection tool(s) and data analysis tool should be stated. Revise your conclusion statement to provide an overall implication of the findings and at least a recommendation.

Introduction

3.The introduction lacks global and regional context. What is the malaria/fever situation globally, regionally, and nationally? What at are the key challenges to malaria/fever globally, regionally, and nationally that have been linked to poor healthcare seeking behaviour? Why is healthcare seeking behaviour in fever cases important?

4.Again, the research problem is unclear and not strongly drafted. The authors need to highlight the impact of poor healthcare seeking behaviour on fever/malaria fight in the country and the district. Why is this district peculiar?

5.Being a qualitative study, this study would have benefitted from a conceptual/theoretical framework. In the results and discussion sections, the authors make references to “individual” “community” and “health facility” level factors. This suggests that the SEM/HUM would best fit the study and its findings in context.

Methods

6.The report should follow the COREQ guidelines for reporting qualitative studies.

7.Revise sentence 1 of paragraph 2 of Study site. The authors use commas instead of points in decimals. These should be rectified.

8.The study design is not stated aside from the authors stating that the approach as qualitative nested in a mixed methods pilot study. Also, why the study design was chosen should be justified. The last sentence is not supposed to be in the study design sub-section but in the procedures.

9.On the population, the authors state two categories of mothers of children under 15 years. This does not feature in the results and discussions as such there is not need for this designation in this sub-section. The justification for the population chosen for this study should be explained in this sub-section. What were the inclusion and exclusion criteria?

10.Rather than the sub-section Recruitment procedures, there should be two subsections, namely; Sampling (where you describe how the different participants were selected and included in the study) and Procedures (which details the data collection procedures including the data collection tools, approaches, and process of data collection). Authors should delete Tables 1 and 2 as they are of little importance if they are able to provide better details of the participants in the Population and Sampling subsections.

11.The meaning of some abbreviations/acronyms used have not be provided in the text. For example, SSI, FGD.

12.How long each interview and FGD took should be stated.

13.In the Abstract, the authors stated that the analysis was thematic. However, in the Data analysis subsection, they state that they used content analysis approach. This should be consolidated and resolved.

Results

14.The first sentence of the sociodemographic characteristics should be sent to the methods section. This is not a result. I also think there should be separate tables for IDI participants and FGD participants.

15.There should a thematic table that summarise the key themes and subthemes identified in the analysis.

16.Separate the subsection for Facilitators and Barriers. This will allow for you to subsection the different levels of factors/barriers and also make it easier for readers’ reading and understanding.

Discussion

17.The first paragraph should restate the specific objectives for the study and the key findings for each which the section discusses.

18.Restructure the discussion to be more coherent and logically flowing following the findings of the specific objectives.

19.The discussion section is deficient in literature comparison, proferring explanations for the study’s observations, and the meaning as well as implications of the findings. Please revise.

Conclusion

20.This section needs revision. The authors only offered to summarise the findings of the study which is not the purpose of this section. Here, the authors should discuss the overall implication of their major findings to policy, practice, and research and based on these, appropriate provide recommendations.

6. PLOS authors have the option to publish the peer review history of their article (what does this mean? ). If published, this will include your full peer review and any attached files.

**Do you want your identity to be public for this peer review?** For information about this choice, including consent withdrawal, please see our Privacy Policy .

Reviewer #1: No

Reviewer #2: **Yes: ** Farrukh Ishaque Saah

---

## [Author Response · Author response to Decision Letter 1]

11 Aug 2024

Response to reviewers

To review 1

Thank you very for all comments about the manuscript. We have addressed all comments, and we believe they improved the manuscript quality presentation. Bellow, are answers of all questions raised.

METHODS: Check use of commas and full stops when presenting numbers e.g. '63.691 inhabitants and 14.583 households' I would write with a comma, not a full stop. Similarly '2,7km (IQR 1,4-7,9 km)' I would write with a full stop, not a comma. This is perhaps a country-specific preference and should be for the editor to make final decision on.

Answer: Thank you for pointing out the error. Based on the recommendations, a full stop was used to write all numbers, while a comma was used to write km.

Some abbreviations used that do not have full description e.g. IQR, NMCP, SSI etc.

Answer: Thank you for the observation, the full description of the indicated abbreviations was inserted.

Line 248 - Confronting who?

Answer: the sentence was rephrased, and the new sentence is now hopefully clearer: it meant to contrast or compare the transcription with the audio recording to verify that they matched accurately.

RESULTS:

Lines 418 - 436: Seems like a lot of important information is included here about different decision-making categories, but is a bit confusing to understand due to long sentences. Can it perhaps be broken down further, or use numbers to signify the different categories?

Answer: The concern about long sentence was addressed in our best capacity. Were broke down some sentences and we rephrased to make it clear.

Line 458: Is women being more 'sensitive' the right word here? Perhaps more aware, or observant, or it needs to be further explained.

Answer: Indeed, the word “sensitive” is from participants’ discourse (“sensível” in the original transcript in Portuguese), and thus we have kept the original word from the participants’ discourse in the quoted text. We indeed think it meant that women may be more attentive and observant to children.

Line 549: What kind of health facility? e.g. health centre, clinic, public, private - perhaps more details about the health facilities can be added in the methods. Additionally, perhaps a few lines on the nature of the health system may help the reader better understand the overall health landscape in Mozambique e.g. are the facilities mainly government funded, or privately run? Are there user fees? Insurance schemes?

Answer: Health facility here refers to the public health centre managed by the government. In this district there is no private health centre at all. We have described the health service available in this study setting in the method section (study setting) and we provided additional sources for more information. Moreover, in Mozambique all uses of public health services are for almost free. They only pay 1 metical (USD 0,016*) for adult consultation and for free for all children, and 5 meticais (USD0,079) for pills, regardless their social status or level of the health facility. Even though, people who do not have this amount still access health services and pills for free. We have added this in the text.

*Rate: USD 1=63,3 (11/0/2024).

Line 782: Do you need to define what a 'formal' healthcare setting is here? If this includes CHWs, perhaps 'service' is better than 'setting'?

Answer: Indeed, the correct word is formal health service. We have corrected it in revised version.

DISCUSSION: Line 953 and 963: The authors could also reference: Virhia J. Contextualising health seeking behaviours for febrile illness: Lived experiences of farmers in northern Tanzania. Health Place. 2022 Jan;73:102710. doi: 10.1016/j.healthplace.2021.102710. Epub 2021 Nov 19. PMID: 34801785 here also in addition to reference 33.

Answer: Thank you for this suggestion. We have included Virhia in the discussion section of the revised manuscript.

Overall, this is a well thought out and implemented research study. The findings are informative and will contribute to the wider literature on health seeking across various contexts. The only key thing I would suggest to add is perhaps providing a diagram or table or description of the overall health landscape in Mozambique identifying who and what the key health services are, whether they are formal/informal/public/private etc. There are many different services and actors mentioned in the study so it may just be useful to have some way to summarise them at the beginning of the results section. See Davis et al, 2021: “If You Do Not Take the Medicine and Complete the Dose…It Could Cause You More Trouble”: Bringing Awareness, Local Knowledge and Experience into Antimicrobial Stewardship in Tanzania - Table 2 (animal health) for a suggestion on how this could be done.

Answer: Thank you for this suggestion. We think that it is a very interesting form of presenting all available health services, but we are lacking data to map all health services in Mozambique, particularly informal ones. The study did map all health services as the article of Davis et al. (2021) did, but rather record only those services emerging from the discourse of the participants. Therefore, we cannot provide a similar table or diagram because we risk to excluded other unknown informal services that did not come out from the data. Moreover, as it was early explained, in Mozambique there is only Public and private health facilities. In urban area there are public and private health services, while in rural areas there are mainly public health services, which we designate health facilities and community health workers. In the study setting there are only 2 formal health services: public health facilities and community health workers. In the text, we have pointed the reader towards a paper on the census of the study district where all public health facilities and community health workers were mapped and described further.

Please check minor spelling errors throughout.

Answer: Thank you very much for warming about it. We have revised the spelling throughout the text.

REFERENCES: Reference 33 should be spelled 'Virhia' not 'Verhia'

Answer: Thank you, error corrected.

To reviewer 2

Thank you very for all comments about the manuscript. We have addressed all comments, and we believe they improved the manuscript quality presentation. Bellow, are answers of all questions raised.

Reviewer #2: Thank you for the change to review this paper. The study’s findings have implication for Mozambique’s malaria elimination fight. However, the manuscript will benefit from major revisions. Find below my comments:

Title & Abstract

1.Delete “Understanding” from the title.

Answer: the new title does not contain the word “understanding”.

2.The Abstract does not present the research problem necessitating the study. “Qualitative approach” is not a study design. The authors should state the exact study design the used for the study. Data collection tool(s) and data analysis tool should be stated. Revise your conclusion statement to provide an overall implication of the findings and at least a recommendation.

Answer: We have revised the abstract to meet the recommendations. However, it was not possible to include all the recommended elements due to the limited word count of 300 words as per PLOS ONE Journal guidelines.

Introduction

3.The introduction lacks global and regional context. What is the malaria/fever situation globally, regionally, and nationally? What at are the key challenges to malaria/fever globally, regionally, and nationally that have been linked to poor healthcare seeking behaviour? Why is healthcare seeking behaviour in fever cases important?

Answer: Further regional and national context on the fever and malaria situation, challenges associated and the implication of care seeking on malaria has been incorporated in the text.

4.Again, the research problem is unclear and not strongly drafted. The authors need to highlight the impact of poor healthcare seeking behaviour on fever/malaria fight in the country and the district. Why is this district peculiar?

Answer: The implication of poor healthcare seeking behaviour for fever on the malaria efforts in this particular district that is unique in that it is undergoing a malaria elimination pilot project has been strengthened in the text. It is important to mention that this study was designed under a malaria elimination research project, and in the text, the reader is pointed towards a number of scientific papers that have been published about it for context, which may answer further questions the reader may have, including:

- Galatas B, Saúte F, Martí-Soler H, Nhamussua L, Simone W, et al. A multiple program for malaria elimination in southern Mozambique (the Magude project): A before-after study. PLoS ONE. 2020; 17(8): e1003227. https://doi.org/10.1371/journal.pmed.1003227

- Galatas B, Nhacolo A, Marti-Soler H, Munguambe K, Jaimise E, Guinovart C, et al. Demographic and health community-based surveys to inform a malaria elimination project in Magude district, Southern Mozambique. BMJ Open. 2020; 10: e033985. https://doi.org/10.1136/bmjopen-2019-033985.

- Galatas B, Nhamtumbo H, Soares R, Djive H, Murato I, Simone W, et al. Community acceptability to antimalarial mass drug administration in Magude district, southern Mozambique: A mixed method study. PLoS ONE. 2021; 16(3): e0249080. https://doi.org/10.1371/journal.pone.0249080

- Portugaliza HP, Galatas B, Nhantumbo H, Djive H, Murato I, Saúte F, Aide P, Pell C, Muguambe K. Examing community perceptions of malaria to inform eliminations efforts in southern Mozambique: a qualitative study. 2019; 18: 232. https://doi.org/101186/s12936-019-2867-y

- Aide P, Candrinho B, Galatas B, Munguambe K, Guinovart C, Luis F, et al. Setting the scene and generating evidence for malaria elimination in Southern Mozambique. Malaria Journal. 2019; 18:190. https//doi.org/10.1186/s12936-019-2832-9 PMID: 31170984.

5.Being a qualitative study, this study would have benefitted from a conceptual/theoretical framework. In the results and discussion sections, the authors make references to “individual” “community” and “health facility” level factors. This suggests that the SEM/HUM would best fit the study and its findings in context.

Answer: We do appreciate this suggestion, we have specified the theoretical orientation: the social ecological model was added as suggested.

Methods

6.The report should follow the COREQ guidelines for reporting qualitative studies.

Answer: we checked the COREQ guidelines for reporting qualitative studies, which was attached within the submission of this manuscript, and we revised this manuscript accordingly. All aspect mentioned in this guideline are inserted in the method. However, we understand that these 32-item check list do not represent the recommend structure of the manuscript as indicated by PLOS ONE. Therefore, the report followed the recommended structure, and considered logically the COREQ guidelines.

7.Revise sentence 1 of paragraph 2 of Study site. The authors use commas instead of points in decimals. These should be rectified.

Answer: Thank you, this sentence was revised, and points were used instead of commas.

8.The study design is not stated aside from the authors stating that the approach as qualitative nested in a mixed methods pilot study. Also, why the study design was chosen should be justified. The last sentence is not supposed to be in the study design sub-section but in the procedures.

Answer: Thank you for the observation, the study design is now specified in the new revised version and the justification was provided.

9.On the population, the authors state two categories of mothers of children under 15 years. This does not feature in the results and discussions as such there is not need for this designation in this sub-section. The justification for the population chosen for this study should be explained in this sub-section. What were the inclusion and exclusion criteria?

Answer: We appreciate the review suggestion about eliminating one of the categories of “mothers”, but actually the study does not use “mother” but women with or without decision-making who were mothers of children under 15 years old, as highlighted in the table of sample size. These two categories were important to analyse healthcare seeking for fever cases as discussed in one of the main themes “health-seeking behaviour”, as the decision-making power highly affected the promptness with which they sought care at formal health services, especially for their children, as stated in the text. In addition, a justification for choosing the included categories was provided. The inclusion criteria were presented Table 1, but as suggested by the reviewer, we deleted the table and we added the inclusion and exclusion criteria in the text.

10.Rather than the sub-section Recruitment procedures, there should be two subsections, namely; Sampling (where you describe how the different participants were selected and included in the study) and Procedures (which details the data collection procedures including the data collection tools, approaches, and process of data collection). Authors should delete Tables 1 and 2 as they are of little importance if they are able to provide better details of the participants in the Population and Sampling subsections.

Answer: We followed most of the recommendations suggested as highlighted in the revised manuscript. However, we think that table 2 is important because it highlight the sample size within different groups of the community. We think that transforming this table in words could create misunderstanding.

11.The meaning of some abbreviations/acronyms used have not be provided in the text. For example, SSI, FGD.

Answer: Thank you, the meaning of all abbreviation/acronyms was provided in the new revised manuscript.

12.How long each interview and FGD took should be stated.

Answer: the duration of interviews and FGD was proved in the last version.

13.In the Abstract, the authors stated that the analysis was thematic. However, in the Data analysis subsection, they state that they used content analysis approach. This should be consolidated and resolved.

Answer: A thematic analysis was used to analyse the data. We have corrected and harmonized the indicated discrepancy in revised version.

Results

14.The first sentence of the sociodemographic characteristics should be sent to the methods section. This is not a result. I also think there should be separate tables for IDI participants and FGD participants.

Answer: We appreciate the suggestion, and we have moved the sociodemographic characteristics of the participants to “sampling” section. The two groups (SSI and FGD) were presented separately.

15.There should a thematic table that summarise the key themes and subthemes identified in the analysis.

Answer: We appreciate the suggestion, but think that a table would represent the content of the result section, which seems not to be recommended by the Journal guidelines. However, to facilitate the reader we have now described the main themes and its sub-themes in text. We hope this will provide a clear understanding of themes and sub-themes emerging from the data.

16.Separate the subsection for Facilitators and Barriers. This will allow for you to subsection the different levels of factors/barriers and also make it easier for readers’ reading and understanding.

Answer: Thank you, the subsection for facilitators and barriers are now separated as shown in the revised version.

Discussion

17.The first paragraph should restate the specific objectives for the study and the key findings for each which the section discusses.

Answer: The specific objectives and the respective study finding are now integrated in the discussion.

18.Restructure the discussion to be more coherent and logically flowing following the findings of the specific objectives.

Answer: Thank you, the new revised version presents a restructured discussion, following the logic of the finding of the specific objectives.

19.The discussion section is deficient in literature comparison, proferring explanations for the study’s observations, and the meaning as well

---

## [Decision Letter · Decision Letter 1]

13 Feb 2025

PONE-D-24-05377R1Healthcare-seeking behaviour of fever cases in Magude district, southern Mozambique: a qualitative studyPLOS ONE

Dear Dr. Cuinhane,

Thank you for submitting your manuscript to PLOS ONE. After careful consideration, we feel that it has merit but does not fully meet PLOS ONE’s publication criteria as it currently stands. Therefore, we invite you to submit a revised version of the manuscript that addresses the points raised during the review process. Please submit your revised manuscript by Mar 30 2025 11:59PM. If you will need more time than this to complete your revisions, please reply to this message or contact the journal office at plosone@plos.org . Please include the following items when submitting your revised manuscript:

We look forward to receiving your revised manuscript.

Kind regards,

Khin Thet Wai, MBBS, MPH, MA

Academic Editor

PLOS ONE

**Journal Requirements:**

**Additional Editor Comments:**

Please do minor revisions as required.

Reviewers' comments:

Reviewer's Responses to Questions

**Comments to the Author**

1. If the authors have adequately addressed your comments raised in a previous round of review and you feel that this manuscript is now acceptable for publication, you may indicate that here to bypass the “Comments to the Author” section, enter your conflict of interest statement in the “Confidential to Editor” section, and submit your "Accept" recommendation.

Reviewer #1: All comments have been addressed

Reviewer #2: All comments have been addressed

2. Is the manuscript technically sound, and do the data support the conclusions?

Reviewer #1: Yes

Reviewer #2: Yes

3. Has the statistical analysis been performed appropriately and rigorously? 

Reviewer #1: N/A

Reviewer #2: N/A

4. Have the authors made all data underlying the findings in their manuscript fully available?

Reviewer #1: Yes

Reviewer #2: Yes

5. Is the manuscript presented in an intelligible fashion and written in standard English?

Reviewer #1: Yes

Reviewer #2: Yes

6. Review Comments to the Author

**Reviewer #1: ** All comments have been satisfactorily addressed by the authors and I now recommend this paper for publication in PLOS ONE

**Reviewer #2:**  Thank you for your revised manuscript. Most of the issues raised have been well-addressed. However, there are still some issues needing revision:

Abstract:

1. Although I understand the concern of the author(s) about the word count for the abstract. However, I still find the abstract deficient. The abstract should clearly summarise why the study, what methods were done, what findings were obtained, and what is the implication of the findings. As such, please revise to ensure the research problem is stated. For instance, was the data analysis done manual or a software was used? This should be stated.

Introduction:

2. This has been improved significantly. However, lines 143-146 are hanging. A single sentence cannot pass as a paragraph. The authors should revise.

Methods:

3. On the population, my previous comment was not to delete a category of mothers. The concern raised was that since the authors argue that this categorization was significant, it expected that it shows in the results and discussions. However, the authors were silent about how this categorization’s peculiar background presents in the findings and the discussion, making this categorization obsolete. Please let this reflect in the details of the quotes and highlight this is in the discussion if relevant findings support this.

Results:

4. I believe socio-demographic information of the study participants should be presented in the results section as they were not predetermined and considered part of the results.

5. I still insist that authors should include a thematic table of the findings. This is not a journal requirement but a good qualitative study practice. It gives a snapshot of the findings and help readers understand the organization and flow of the themes and their subthemes. Please provide this.

6. Rather than quoting even the researcher’s questions in the quotes, it is best practice to only quote that of the participant and add any context/explanation in parenthesis. For instance, lines 865-872 should best be presented as: “Today, I have a cough. It started yesterday. …I made syrup. …(no need to go to health facilty because) it is still not strong. …Yes (it is able to treat oneself at home).”

Discussion:

7. This has improved a lot. However, it is not enough just stating that a finding is similar or different from that of previous studies. Authors should offer at least a possible explanation for difference in findings.

8. The subsection limitations should instead be Strengths and Limitations to also espouse the strengths of the study and not only the limitations.

7. PLOS authors have the option to publish the peer review history of their article (what does this mean? ). If published, this will include your full peer review and any attached files.

**Do you want your identity to be public for this peer review?** For information about this choice, including consent withdrawal, please see our Privacy Policy .

Reviewer #1: No

Reviewer #2: No

---

## [Author Response · Author response to Decision Letter 2]

26 Mar 2025

1. Response to Editor

We have reviewed the reference list to ensure that it is complete and correct. None of the papers cited have been retracted. Some editorial changes have been made for typos found in some of the references’ names.

2. Response to reviewers

Abstract:

1. Although I understand the concern of the author(s) about the word count for the abstract. However, I still find the abstract deficient. The abstract should clearly summarise why the study, what methods were done, what findings were obtained, and what is the implication of the findings. As such, please revise to ensure the research problem is stated. For instance, was the data analysis done manual or a software was used? This should be stated.

Answer: Thank you for guiding us on how to make the abstract more impactful. We have revised the abstract to make sure the elements recommended are presented, including the research question necessitating the study. We have also included the methods used, the software used for data management, as well as a summary of the main findings and their implications and recommendations.

Introduction:

2. This has been improved significantly. However, lines 143-146 are hanging. A single sentence cannot pass as a paragraph. The authors should revise.

Answer: Thank you very much for this observation. The issue has been addressed and the lines are no longer hanging, and language has been revised in the introduction.

Methods:

3. On the population, my previous comment was not to delete a category of mothers. The concern raised was that since the authors argue that this categorization was significant, it expected that it shows in the results and discussions. However, the authors were silent about how this categorization’s peculiar background presents in the findings and the discussion, making this categorization obsolete. Please let this reflect in the details of the quotes and highlight this is in the discussion if relevant findings support this.

Answer: Thank you for the observation. We have reviewed the representation of these categories in the results and discussion sections and made some relevant changes to present the specific idiosyncrasies associated to health care seeking behaviors in families where women have decision making powers in contrast to those where they don’t, showing that women with decision making power promptly seek care for children. We have realized that despite including relevant quotes from women with decision-making power, community leaders (which is the group of man who specifically stated women in their households hold the decision-making power in relation to child care), and women without decision-making power, we needed to make the relevance of the quotes explicit in the text in order for this difference to be better displayed. We have now done so within results, in the section on decision-making processes and timing to seek care (lines 483-517). Because the decision-making power held by women, who are the main caretakers of children in this community, affects the promptness with which care is sought in this study and other studies, this has been discussed in the discussion in lines 1058-1093. It is important to explain that the “category of mother” was not labelled as such in this study, but a characteristic of a category designated “women with or without decision making”, who are mothers of children under 15 years old, as presented in Table 1. Based on this understanding, we have reviewed the representation of these categories (women with decision making and women without decision-making) in the results and discussion sections.

Results:

4. I believe socio-demographic information of the study participants should be presented in the results section as they were not predetermined and considered part of the results.

Answer: We have welcomed the observation and moved the socio-demographic information to the results section as suggested.

5. I still insist that authors should include a thematic table of the findings. This is not a journal requirement but a good qualitative study practice. It gives a snapshot of the findings and help readers understand the organization and flow of the themes and their subthemes. Please provide this.

Answer: We agree that this will help he reader have a snapshot of the main themes and sub-themes in the study and thus a thematic table of findings has now been included in the results section, thank you for the suggestion.

6. Rather than quoting even the researcher’s questions in the quotes, it is best practice to only quote that of the participant and add any context/explanation in parenthesis. For instance, lines 865-872 should best be presented as: “Today, I have a cough. It started yesterday. …I made syrup. …(no need to go to health facility because) it is still not strong. …Yes (it is able to treat oneself at home).”

Answer: Thank you very much for pointing this out. We have corrected the presentation of quotes to show the participants’ answers only in quotes that previously contained researcher’s questions (lines 386-390; 516-517; 744-747; 810-812).

Discussion:

7. This has improved a lot. However, it is not enough just stating that a finding is similar or different from that of previous studies. Authors should offer at least a possible explanation for difference in findings.

Answer: Thank you, this suggestion enrichens the relevance of the study in its specific context. We have now provided possible explanations for contrasting findings as well as for some findings aligned with other studies, in lines 1027-1032, 1051-1056, 1116-1119, and 1169-1179 of the discussion.

8. The subsection limitations should instead be Strengths and Limitations to also espouse the strengths of the study and not only the limitations.

Answer: Thank you very much for pointing this out. We have course corrected and included study strengths in lines 1229-1238 as part of the “Strengths and limitations” section.

---

## [Decision Letter · Decision Letter 2]

3 Apr 2025

Healthcare-seeking behaviour of fever cases in Magude district, southern Mozambique: a qualitative study

PONE-D-24-05377R2

Dear Dr. Cuinhane,

We’re pleased to inform you that your manuscript has been judged scientifically suitable for publication and will be formally accepted for publication once it meets all outstanding technical requirements.

Kind regards,

Khin Thet Wai, MBBS, MPH, MA

Academic Editor

PLOS ONE

Additional Editor Comments (optional):

Reviewers' comments:

Reviewer's Responses to Questions

**Comments to the Author**

1. If the authors have adequately addressed your comments raised in a previous round of review and you feel that this manuscript is now acceptable for publication, you may indicate that here to bypass the “Comments to the Author” section, enter your conflict of interest statement in the “Confidential to Editor” section, and submit your "Accept" recommendation.

Reviewer #2: All comments have been addressed

2. Is the manuscript technically sound, and do the data support the conclusions?

Reviewer #2: Yes

3. Has the statistical analysis been performed appropriately and rigorously? 

Reviewer #2: N/A

4. Have the authors made all data underlying the findings in their manuscript fully available?

Reviewer #2: Yes

5. Is the manuscript presented in an intelligible fashion and written in standard English?

Reviewer #2: Yes

6. Review Comments to the Author

Reviewer #2: All the issues have been address satisfactorily.

Just a minor comment: In the parenthesis of the participants' details after the quote presentation, there is a grammatical issue of using a plural noun rather than a singular one. For example, “men, household head” and “women with decision-making power” instead of “man, household head” and “woman with decision-making power”

Read through and correct these.

7. PLOS authors have the option to publish the peer review history of their article (what does this mean? ). If published, this will include your full peer review and any attached files.

**Do you want your identity to be public for this peer review?** For information about this choice, including consent withdrawal, please see our Privacy Policy .

Reviewer #2: **Yes: ** Farrukh Ishaque Saah

---

## [Editor Report · Acceptance letter]

PONE-D-24-05377R2

PLOS ONE

Dear Dr. Cuinhane,

I'm pleased to inform you that your manuscript has been deemed suitable for publication in PLOS ONE. Congratulations! Your manuscript is now being handed over to our production team.

Kind regards,

on behalf of

Dr. Khin Thet Wai

Academic Editor

PLOS ONE